# Asymmetric sheath coordination controls flagellar architecture and function in *Leptospira* spirochete

Akihiro Kawamoto [ID][1][✉], Toshiki Kuribayashi[2], Masatomo Morita [ID][3], Shuichi Nakamura [ID][2][✉] & Nobuo Koizumi [ID][3][✉]

## Abstract

**Bacterial flagella are essential for motility, but their structure and how they generate movement vary greatly. Most motile bacteria use external helical flagella, whereas spirochetes have periplasmic flagella (PFs) that distort the cell body to drive forward movement. Here, we generated sheath protein knockout mutants and used high-resolution cryo-electron microscopy to elucidate the mechanisms underlying PF assembly, curvature, and rigidity in *Leptospira biflexa*. The PF consists of a FlaB1-based core filament surrounded asymmetrically by sheath proteins. Weak but essential binding of FlaA2 to the core enables asymmetric localization of the coiling protein FcpA. FcpA alone can induce curvature, whereas FcpB acts as a structural wedge that reinforces PF rigidity and enables efficient swimming in liquid. Specific glycosylation of FlaB1 mediates sheath-core interactions and may guide the assembly of sheath components. We propose that sheath proteins interact transiently with the core and may be anchored to the outer membrane, allowing core rotation beneath a static sheath. These findings reveal how cooperative interactions among sheath components confer structural and mechanical specialization to spirochete flagella.**

**Keywords** Bacterial Flagella; Bacterial Motility; Cryo-Electron Microscopy; *Leptospira*; Spirochete
**Subject Categories** Microbiology, Virology & Host Pathogen Interaction; Structural Biology

## Introduction

The bacterial flagellum is a supramolecular machine whose rotation enables locomotion in liquid or over surfaces. Flagellum-dependent motility is critical for navigating toward favorable conditions and contributes to virulence in many pathogenic species (Nakamura and Minamino, 2019). The flagellum consists of a basal body (the flagellar motor) and a helical filament (the flagellar filament), which functions as a rotary engine and a screw-like propeller, respectively. While the fundamental structure and function of the flagellar motor are highly conserved across species, variations exist in its energy source—such as proton or sodium ions—and in its maximum rotational speed. In addition to torque generation, the flagellar motor also serves as a protein export apparatus and is classified as a type III secretion system, sharing partial sequence similarity and structural and functional features with other injectosome systems, including the *Salmonella* needle complex and *Bacillus subtilis* sporulation system (Kawamoto et al, 2013; Johnson et al, 2020). In contrast, the flagellar filament, which is essential for generating thrust, is a flagellum-specific component and exhibits considerable diversity among bacterial species. In terms of protein composition, some species form flagella using a single type of flagellin (e.g., *Escherichia coli* and *Salmonella enterica*), while others utilize multiple flagellins (e.g., six in *Caulobacter crescentus*, seven in *Rhizobium leguminosarum*, and two in *Campylobacter jejuni* and *Shewanella putrefaciens*) (Nakamura and Minamino, 2019). In species with multiple flagellin genes, some flagellins exhibit functional redundancy; flagella can still assemble and support motility, albeit sometimes with morphological abnormalities, even when one or more flagellins are absent (Faulds-Pain et al, 2011; Li et al, 2008). For instance, wild-type (WT) *C. crescentus* expresses FljJ, FljK, FljL, FljM, FljN, and FljO, yet flagellar assembly and motility persist even when five of these six proteins are deleted (Faulds-Pain et al, 2011). Spirochetes are characterized by the presence of endoflagella—flagella located within the periplasmic space between the outer and cytoplasmic membranes—also known as periplasmic flagella (PFs). The PFs of *Brachyspira hyodysenteriae*, a spirochete pathogen that causes swine dysentery, consist of a core filament composed of FlaB1, FlaB2, and FlaB3, surrounded by an outer sheath made of FlaA. While the FlaB proteins can compensate for the loss of one another, deletion of FlaA leads to loss of the sheath and alters flagellar helicity (Li et al, 2008).

Functional compensation between flagellar proteins may serve as a safeguard to prevent the complete loss of flagella, which are essential for bacterial survival and infection, and this phenomenon is of evolutionary interest. At the same time, the distinct roles of individual flagellar components, such as FlaA and FlaB in *B. hyodysenteriae*, confer species-specific properties to the flagellum.

[1]Institute for Protein Research, The University of Osaka, 3-2 Yamadaoka, Suita, Osaka 565-0871, Japan. [2]Department of Applied Physics, Graduate School of Engineering, Tohoku University, 6-6-05 Aoba, Aoba, Sendai, Miyagi 980-8579, Japan. [3]Department of Bacteriology I, National Institute of Infectious Diseases, Japan Institute for Health Security, 1-23-1 Toyama, Shinjuku, Tokyo 162-8640, Japan. ✉E-mail: kawamoto@protein.osaka-u.ac.jp; shuichi.nakamura.e8@tohoku.ac.jp; koizumi.n@jihs.go.jp

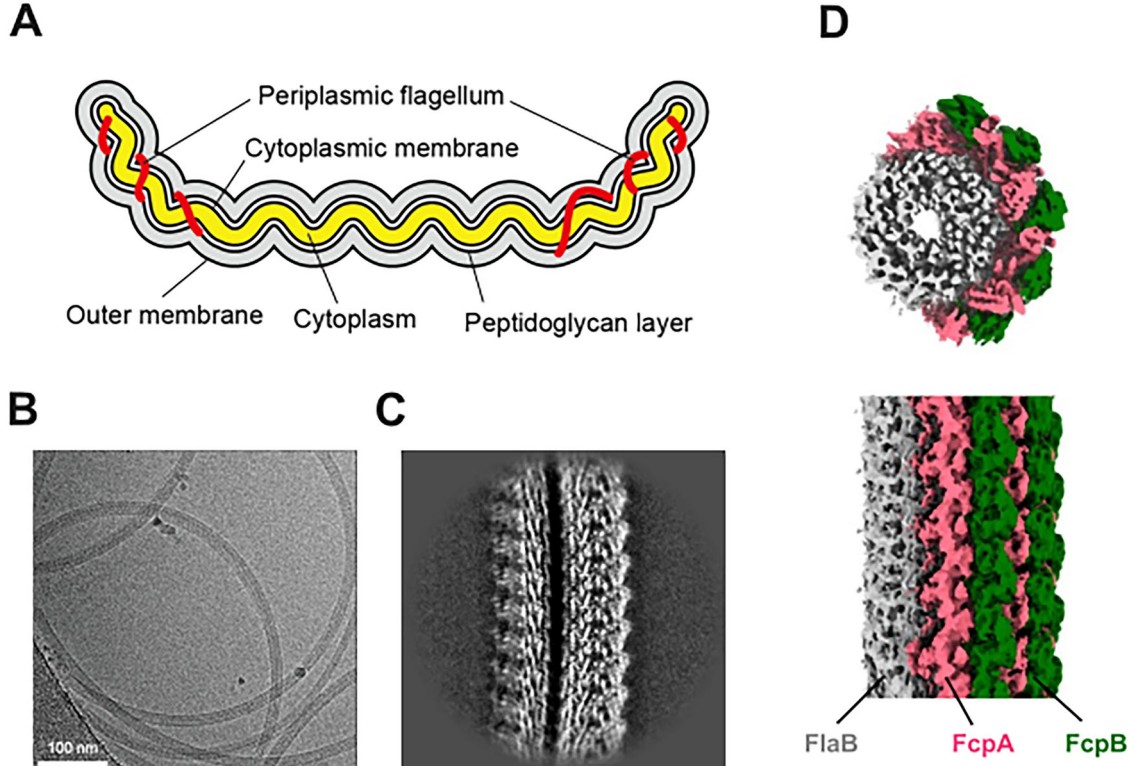

**Figure 1. Structure of wild-type *Leptospira* PFs.**

(A) Schematic diagram of a *Leptospira* cell, illustrating the localization of the periplasmic flagella between the outer and cytoplasmic membranes. (B) A cryo-EM micrograph of purified PFs from WT strain showing their coiled morphology. (C) Averaged 2D class image of WT obtained by single-particle analysis. (D) Reconstructed 3D structure of the coiled WT PF. The core filament is composed of FlaB (gray), asymmetrically surrounded by the sheath proteins FcpA (magenta) and FcpB (green). See Methods for details of reconstruction reproducibility. Source data are available online for this figure.

Understanding these mechanisms can provide insights into the diversity of the bacterial motility system. Among bacteria with flagella composed of multiple types of proteins, the spirochete *Leptospira* spp.—whose pathogenic strains cause the widespread zoonotic disease leptospirosis—represents a notable exception, as the loss of individual components can result in distinct, non-redundant phenotypes. *Leptospira* spp. possess one PF at each end of the cell body (two PFs per cell). These PFs bend the ends of the cell into characteristic hook shapes (Fig. 1A) (Bromley and Charon, 1979; Nakamura, 2020). Their structural organization, comprising a core filament and an outer sheath, resembles that of *B. hyodysenteriae*. The core filament is formed by FlaB proteins, while the sheath is currently believed to include at least four proteins: FlaA1 and FlaA2, which are also found in *B. hyodysenteriae*, and FcpA (flagellar coiling protein A) and FcpB, which are unique to *Leptospira* spp. (Wunder et al, 2016; Sasaki et al, 2018; Wunder et al, 2018; Gibson et al, 2020). Lambert et al generated *L. interrogans* mutants lacking FlaA1, or both FlaA1 and FlaA2, via random transposon mutagenesis (Lambert et al, 2012). The double mutant lacking FlaA1 and Fla2 exhibited neither hook-shaped cell ends nor translational motility and was avirulent in a hamster model of acute infection. In contrast, the single *flaA1::Tn* mutant retained normal morphology and virulence but showed reduced motility. The PFs from the double mutant had a thickness similar to that of the WT PFs, suggesting the presence of additional sheath

proteins. FcpA was later identified through the analyses of a clinical *L. interrogans* isolate and a *L. biflexa* transposon mutant (Wunder et al, 2016; Sasaki et al, 2018; Wunder et al, 2018; Gibson et al, 2020). PFs from the *ΔfcpA* mutants were thinner than those of the WT and exhibited a straight conformation, leading to the loss of hook-shaped cell ends. FcpB was subsequently identified as another sheath protein contributing to the hook-shaped morphology, the coiled structure of PFs, and high motility (Wunder et al, 2018; Gibson et al, 2020). Cryo-electron microscopy (EM) revealed the localization of FcpB on the outer side of the curved flagellar filament (Wunder et al, 2018; Gibson et al, 2020).

During our analysis of PFs from the *ΔfcpA* mutant (Sasaki et al, 2018), we identified a protein that was diminished in the purified FcpA-deficient PFs. Tandem mass spectrometry revealed this protein to be FcpB (Appendix Fig. S1; Appendix Table S1). To further investigate the role of FcpB, we generated *fcpB* knockout mutants via allelic exchange and observed distinct colony sizes on semi-solid agar plates (Appendix Fig. S2). While one mutant, in which the *fcpB* gene was disrupted by a spectinomycin resistance cassette as intended, produced colonies similar in size to the WT, another clone displayed less motility. Whole genome sequencing revealed that this less motile clone carried a single deoxyguanosine insertion in the *flaA2* gene, causing a frameshift at amino acid position 110 and abolishing FlaA2 expression (Appendix Fig. S3). These results highlight the importance of FlaA2 for motility in *L.*

*biflexa*. Collectively, our findings suggest that sheath proteins contribute to flagellar curvature, yet whether they act cooperatively or perform distinct roles remains unresolved. In this study, we aimed to dissect the individual contributions of the sheath proteins in *Leptospira* PFs. We constructed knockout mutants of *fcpB, flaA1*, and *flaA2*, and analyzed their effects on flagellar morphology and mechanical properties. By integrating these phenotypic analyses with structural insights from cryo-EM, we propose distinct and cooperative roles for each sheath protein in shaping the unique architecture of the *Leptospira* flagellum.

# Results

## PF structures of the WT strain of *L. biflexa*

Purified PFs from the WT strain of *L. biflexa* were imaged by cryo-EM (Fig. 1B,C), and a structure with a resolution of 3.24 Å was obtained (Fig. EV1). This structure revealed the asymmetric localization of sheath proteins on the outer curvature of the FlaB core filament, consistent with a previous report (Gibson et al, 2020) (Fig. 1C,D). Based on homology models derived from mutant strains (see below), the sheath protein adjacent to the FlaB core was identified as FcpA, while the outermost sheath protein was identified as FcpB. Although FlaA2 was detected in purified WT PFs by immunoblotting (Sasaki et al, 2018), no corresponding density was observed in the single-particle reconstruction (Fig. 1D), suggesting that FlaA2 may have dissociated during purification or was disordered in this structural state.

## FlaA2, but not FlaA1, is required for the formation of coiled PFs

*L. biflexa* contained two *flaA* genes, *flaA1* and *flaA2*, arranged in an operon with *flaA2* located upstream of *flaA1*, as seen in *L. interrogans* (Picardeau et al, 2008). To investigate the roles of FlaA proteins in the assembly of coiled *Leptospira* PF, we generated Δ*flaA1* and Δ*flaA2* knockout strains by homologous recombination. Because the two genes are arranged in an operon, the Δ*flaA2* strain lacks expression of both proteins (Fig. EV2). On semi-solid agar plates, the Δ*flaA2* mutant formed significantly smaller colonies than the WT strain (Figs. 2A, top and EV2B), whereas deletion of *flaA1* alone did not affect the colony size (Appendix Fig. S4). As expected from its reduced colony size, the swimming speed of the Δ*flaA2* strain was significantly slower than that of the WT strain ($P = 1.16 \times 10^{-5}$; Fig. 2B). Dark-field microscopy revealed that the Δ*flaA2* cells exhibited straight cell ends (Fig. 2A, middle), and negative staining transmission electron microscopy showed that their PFs lacked the characteristic coiled morphology (Fig. 2A, bottom). This straight PF phenotype was also confirmed by dark-field imaging of isolated PFs in liquid (Appendix Fig. S5). These results indicate that the loss of coiled PFs underlies the straightened cell shape and reduced motility of the Δ*flaA2* mutant. Immuno-blotting confirmed that FcpA and FcpB were still expressed in the Δ*flaA2* mutant (Fig. 2C). Cryo-EM analysis at 2.32 Å resolution revealed that these proteins form a uniform sheath over the FlaB core filament (Fig. 2D; Appendix Fig. S6), in contrast to the asymmetric distribution seen in WT (Fig. 1D). Although FlaA1 was previously detected in PFs using a different extraction method

(Sasaki et al, 2018), repeating that method failed to detect FlaA1, suggesting that FlaA1 is not stably associated with the core filament and/or other sheath proteins, and may have dissociated during purification (Appendix Fig. S7). Introduction of *flaA2* alone into the Δ*flaA2* mutant restored cell shape, PF morphology, motility and the asymmetric localization of FcpA and FcpB (Fig. 2A,B,E). The colony size of the complemented strain was smaller than that of the WT, likely due to slower growth in the presence of kanamycin, as its swimming speed was comparable to that of the WT (Figs. 2B and EV2). Notably, additional densities (colored orange) were observed on the outer edge of the sheath in the complemented strain (Fig. 2E; Appendix Fig. S8) but not in WT (Fig. 1D). Although these structures could not be conclusively identified due to limited resolution, they differed from FcpA and FcpB and are therefore likely to represent FlaA2. The streaky densities observed in these reconstructions may, at least in part, reflect curvature heterogeneity of the filaments. These findings indicate that FlaA2 is essential for organizing the asymmetric sheath structure by directing the proper localization of FcpA and FcpB, thereby enabling the formation of coiled PFs. The potential role of FlaA2 in guiding sheath assembly is further discussed below.

## FcpB stabilizes FcpA structure

Due to the inherent curvature of WT PFs, obtaining high-resolution cryo-EM reconstructions is challenging. To enable high-resolution structural analysis, we therefore examined straight PFs from the Δ*flaA2* and Δ*fcpB*/*flaA2* strains. The 2.37 Å resolution structure of PFs from the Δ*fcpB*/*flaA2* strain, along with the PF structure from the Δ*flaA2* strain described above, showed that FcpA and FcpB formed separate rows, each composed exclusively of FcpA or FcpB, respectively (Figs. 2D and 3A,B). The FcpB fitted snugly between adjacent FcpA rows (Figs. 3B and EV3), and electrostatic surface potential maps suggested that electrostatic interactions contribute to the association between the two proteins (Fig. 3C,D). A comparison of the FcpA structure from the FcpB-deficient Δ*flaA2* strain with the density map of FcpA interacting with FcpB showed that insertion of FcpB did not affect the width of the FcpA rows (Fig. EV4). Structural analysis of FcpA from FcpB-deficient PFs revealed that the N-terminal region was not well resolved (highlighted in yellow in Fig. 3E). In contrast, in the structure derived from the Δ*flaA2* strain, in which the sheath is composed of both FcpA and FcpB, the N-terminal residues of FcpA were clearly defined (Fig. 3F), suggesting that FcpB stabilizes the N-terminal structure of FcpA. This interaction between FcpA and FcpB was further supported by an immunoprecipitation assay (Appendix Fig. S9).

## FcpB is required for efficient motility under low-viscosity conditions

The Δ*fcpB*_CL13 strain, constructed via allelic exchange, formed significantly smaller colonies than the WT strain on 0.4% agar plates ($P = 2.73 \times 10^{-77}$; Fig. 4A,B). This phenotype, consistent with the reduced colony spreading reported by Wunder et al, (2018), is attributable to the loss of FcpB, as confirmed by immunoblotting of purified PFs from the mutant (Fig. 4C). In viscous medium, Wunder et al observed only a slight motility defect upon *fcpB* deletion; in agreement, our Δ*fcpB*_CL13 strain swam somewhat

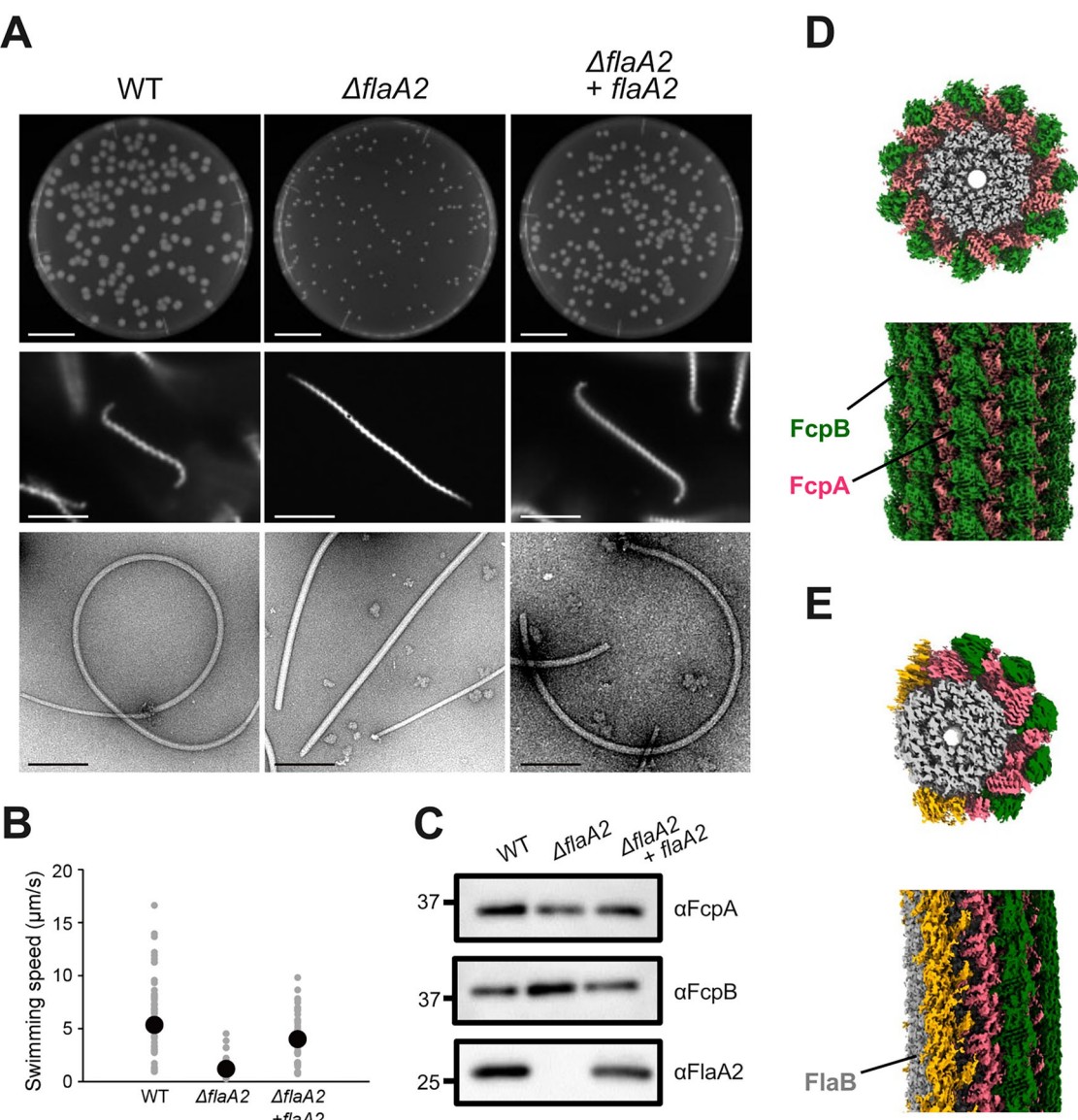

**Figure 2. FlaA2 is essential for functional PFs by regulating the localization of FcpA and FcpB.**

(A) Phenotypes of WT, *flaA2*-knockout (*ΔflaA2*), and *flaA2*-complemented *ΔflaA2* strains. Upper panels: colony morphology on 0.4% soft agar plates after 7 days of incubation at 30 °C. The assay was performed twice independently, and representative images are shown. Bar = 2 cm. The WT colony image is reused in Appendix Fig. 4B. Middle panels: dark-field microscopy images of cells. Bar = 5 μm. Lower panels: transmission electron microscopy images of negatively stained isolated PFs. Bar = 200 nm. (B) Swimming speeds of WT, *ΔflaA2*, and *flaA2*-complemented strains. All individual data points are plotted, with mean values indicated by black circles. Data were collected from 26 WT, 53 *ΔflaA2*, and 48 *flaA2*-complemented cells, obtained from two biological replicates. Statistical significance was assessed using a two-tailed Student's *t*-test, assuming unequal variances (WT vs *ΔflaA2* $P = 1.16 \times 10^{-5}$, WT vs *ΔflaA2 + flaA2* $P = 0.12$, *ΔflaA2* vs *ΔflaA2 + flaA2* $P = 2.91 \times 10^{-10}$). (C) Immunoblot analysis of purified PFs from WT, *ΔflaA2*, and *flaA2*-complemented strains, using antisera against FcpA, FcpB, and FlaA2. PFs were prepared in two independent experiments, and each preparation was analyzed by immunoblotting; a representative blot is shown. (D) Cryo-EM reconstruction of PFs from *ΔflaA2* strain. The FlaB core (gray) is uniformly surrounded by FcpA (magenta) and FcpB (green). (E) Cryo-EM reconstruction of PFs from *flaA2*-complemented strain. The FlaB core (gray) is asymmetrically covered by FcpA (magenta) and FcpB (green). Additional densities (orange) on the outer sheath are presumed to be FlaA2. See Methods for details of reconstruction reproducibility. Source data are available online for this figure.

more slowly than WT in 0.5% methylcellulose (MC), swimming at ~70% of the WT speed (*P* = 0.18; Fig. 4D). The soft-agar motility assay likewise revealed reduced motility under high-viscosity conditions (Appendix Fig. S10). Growth analysis further showed that the *ΔfcpB*_CL13, as well as another *ΔfcpB* mutant (*ΔfcpB*_CL15), grew modestly more slowly than the WT strain (Appendix Fig. S2). Together with the reduced swimming speed,

this slower growth may have contributed to the reduced colony size and spreading of the *ΔfcpB* mutant on 0.4% agar. Both WT and *ΔfcpB*_CL13 strains exhibited higher swimming speeds in 0.5% MC compared with 0% MC (Fig. 4D). In contrast, *ΔfcpB*_CL13 displayed markedly reduced motility in motility buffer without MC (0% MC), swimming at ~15% of the WT speed (*P* = 3.62 × $10^{-13}$; Fig. 4D). Thus, while *ΔfcpB*_CL13 retained its ability to

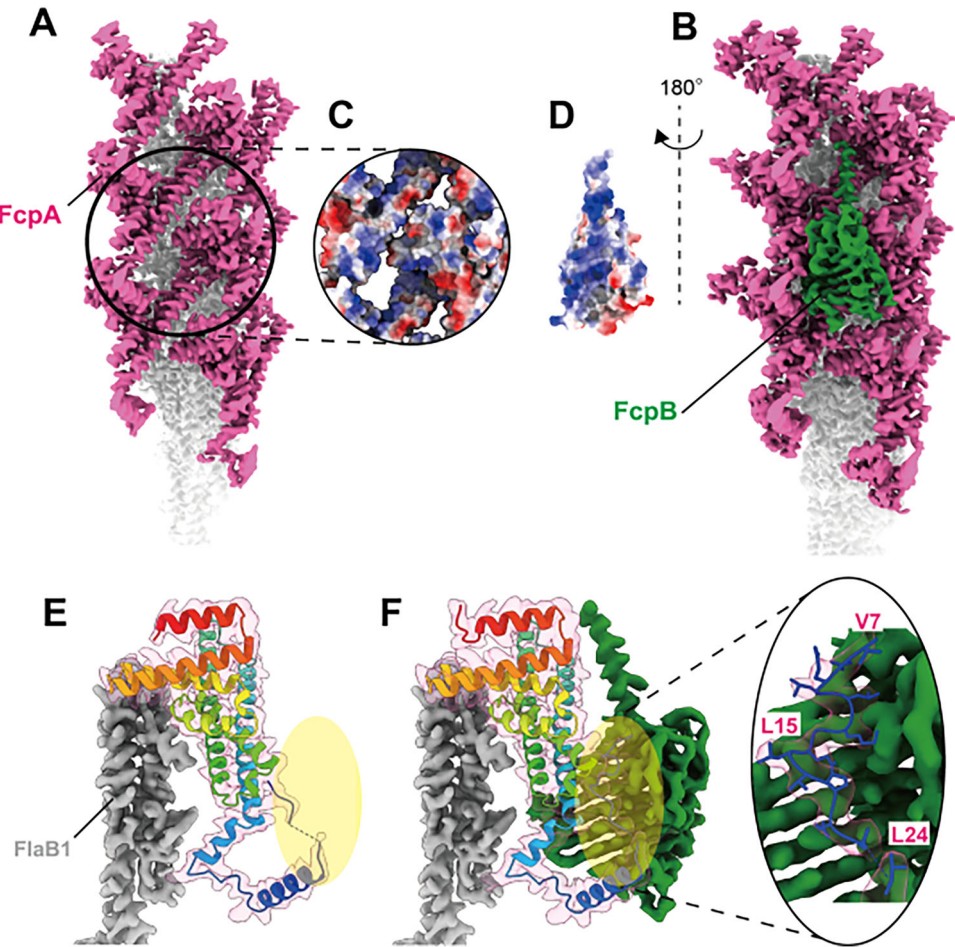

**Figure 3. Interaction between the sheath proteins FcpA and FcpB.**

(A) Reconstructed image of the PF from ΔfcpB/flaA2 strain. The FlaB core filament (gray) is covered only by the sheath protein FcpA (magenta). (B) Reconstructed image of the PF from ΔflaA2 strain. The FlaB core filament (gray) is covered by the sheath proteins FcpA (magenta) and FcpB (green). (C) Electrostatic potential map of the region circled in (A). (D) Electrostatic potential map of FcpB, rotated 180° relative to the view circled in (B). (E) Fitted model of the FlaB core filament (gray) and FcpA (rainbow ribbon model with density map) of the PF from ΔfcpB/flaA2 strain (corresponding to A). (F) Fitted model of the FlaB core filament (gray), FcpA (rainbow ribbon model with density map), and FcpB (green) of the PF from ΔflaA2 strain (corresponding to B), highlighting the N-terminal region of FcpA (superimposed view). See Methods for details of reconstruction reproducibility.

respond to increased viscosity, its motility was severely impaired under low-viscosity conditions, highlighting that FcpB is critical for efficient motility under low-viscosity conditions.

## FcpB increases PF stiffness

The cell ends of the ΔfcpB mutant appeared nearly straight but still exhibited a slight curvature (Fig. 5A). Dark-field microscopy revealed that PFs purified from the ΔfcpB mutant exhibited a coiled shape similar to that of those from the WT (Fig. 5B). Cryo-EM analysis of the mutant PFs revealed that FcpA was still asymmetrically located on the outer surface of the curved filament, indicating that FcpA incorporation does not depend on FcpB (Fig. 5C; Appendix Fig. S11). Given the diminished curvature at the cell ends in the ΔfcpB mutant ($P = 2.21 \times 10^{-10}$; Fig. 5A,D), we hypothesized that the absence of FcpB reduces PF stiffness to a level insufficient to bend the cell ends. To test this, we measured the curvature of isolated PFs from WT and ΔfcpB strains. The ΔfcpB

PFs exhibited a curvature that was slightly but significantly reduced compared to the WT ($P = 2.36 \times 10^{-5}$; Fig. 5D). Using linear elastic theory (see Methods), we estimated the relative stiffness of PFs to the cell body. The stiffness ratio for WT PFs ($A_{WT-PF}/A_{Cell}$) was ~0.3, in good agreement with theoretical predictions (~0.2) by Kan and Wolgemuth (2007) (Fig. 5E). In contrast, the stiffness ratio for ΔfcpB PFs was ~0.06, indicating that WT PFs are about five times stiffer than ΔfcpB PFs. These results suggest that FcpB plays a critical role in conferring stiffness to the PF, enabling it to exert sufficient force to bend the cell ends and maintain the characteristic hook-shaped morphology of *L. biflexa* (Fig. 5F).

## Structural basis of core filaments and their interactions with sheath proteins

In our previous study, both FlaB1 and FlaB2 were detected in PFs purified from the WT strain of *L. biflexa* (Sasaki et al, 2018). However, as described above, the PFs from the WT strain are

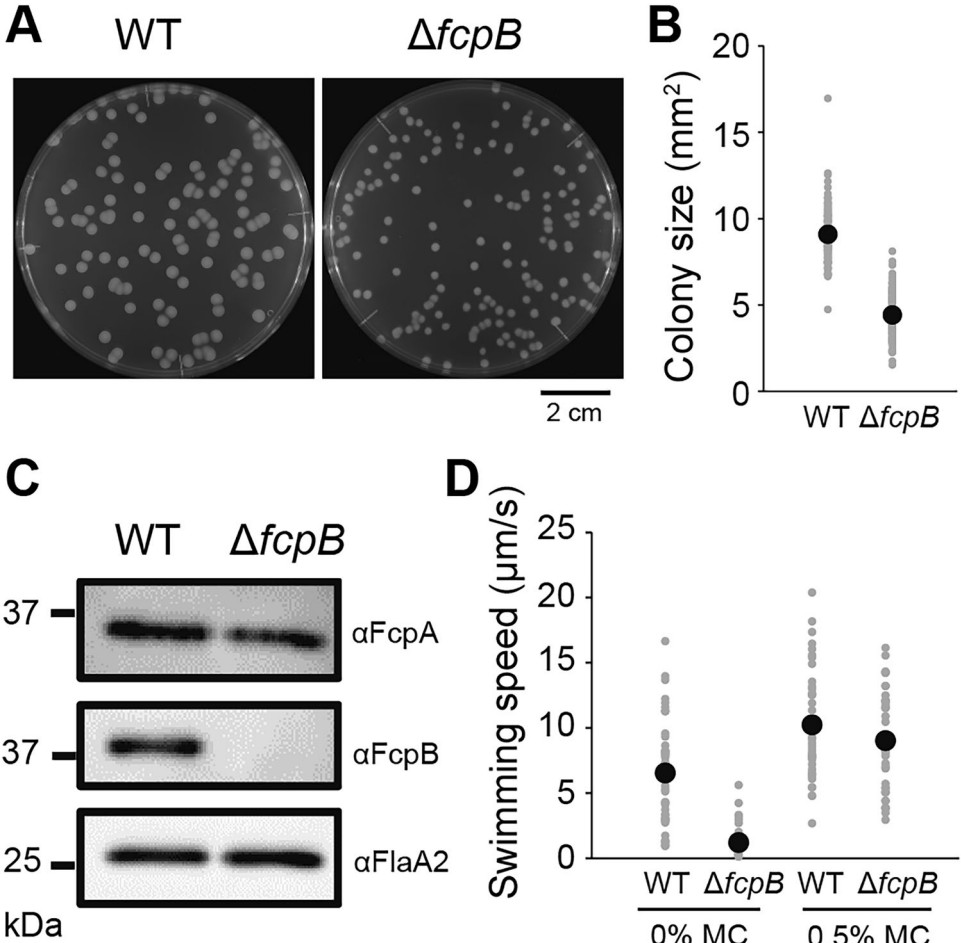

**Figure 4. FcpB deficiency causes pronounced motility defects under low-viscosity conditions.**

(A) Colony morphology of WT and Δ*fcpB* (Δ*fcpB*_CL13) strains on 0.4% soft agar plate after 7 days of incubation at 30 °C. The assay was performed twice independently, and representative images are shown. The WT colony image is reused in Appendix Fig. 2A. (B) Quantification of colony size of WT and Δ*fcpB*_CL13 strains. All individual data points are plotted, with mean values indicated by black circles. Data were collected from 131 colonies for WT and 147 colonies for Δ*fcpB*_CL13, obtained from two independent biological replicates. Statistical significance was assessed using Student's *t*-test ($P = 2.73 \times 10^{-77}$). (C) Immunoblot analysis of PFs purified from WT and Δ*fcpB*_CL13 strains using antisera against FcpA, FcpB, and FlaA2. PFs were prepared in two independent experiments, and each preparation was analyzed by immunoblotting; a representative blot is shown. The immunoblot images probed with anti-FcpA and anti-FcpB are reused in Appendix Fig. 2C. (D) Swimming speeds of WT and Δ*fcpB*_CL13 strains in standard motility buffer (0% methylcellulose, MC) and in buffer containing 0.5% MC. All individual data points are plotted, with mean values indicated by black circles. Data were collected from 48 WT and 49 Δ*fcpB*_CL13 cells in 0% MC, and 40 WT and 40 Δ*fcpB*_CL13 cells in 0.5% MC, obtained from two biological replicates. Statistical significance was assessed using Student's *t*-test (WT vs Δ*fcpB* in 0% MC $P = 3.62 \times 10^{-13}$, WT vs Δ*fcpB* in 0.5% MC $P = 0.18$). Source data are available online for this figure.

highly curved, which prevented the acquisition of high-resolution cryo-EM structures and thus precluded the identification of the FlaB isoform constituting the core filaments. In contrast, the PFs purified from the Δ*fcpB*_KO15 strain (sheathed uniformly with FcpA) and the Δ*flaA2* strain (sheathed uniformly with both FcpA and FcpB) are straight, enabling the acquisition of high-resolution structures (Appendix Figs. S6 and S12). These structures allowed unambiguous identification of the FlaB isoform forming the core filaments. Interestingly, unsheathed core filaments in both strains were composed solely of FlaB2, whereas sheathed filaments consisted exclusively of FlaB1 (Appendix Figs. S6 and S12). Glycosylation of flagellar filaments has been reported in various bacteria, including spirochetes (Li et al, 1993; Brimer and Montie, 1998; Wyss, 1998; Thibault et al, 2001). In our study, we observed

additional electron densities at several serine and threonine residues on FlaB1—specifically $Ser_{115}$, $Thr_{137}$, $Ser_{148}$, $Ser_{187}$, $Thr_{167}$, and $Thr_{182}$—suggesting that these densities likely represent glycans (Fig. 6A,B,C; Appendix Fig. S13). These putative glycans, attached to defined serine and threonine residues on FlaB1, are positioned at the interface between the core filament and sheath proteins, indicating their potential role in mediating core–sheath interactions. In addition, we detected unassigned electron densities (indicated by an asterisk in Fig. 6C) at the interface between the FlaB1 core filament and FcpA, which may also represent glycans. In the absence of these putative glycans, the cryo-EM map revealed a gap between the core and sheath, indicating that direct amino acid interactions between core and sheath are minimal (Fig. 6B). Taken together, these observations suggest that glycosylation of FlaB1

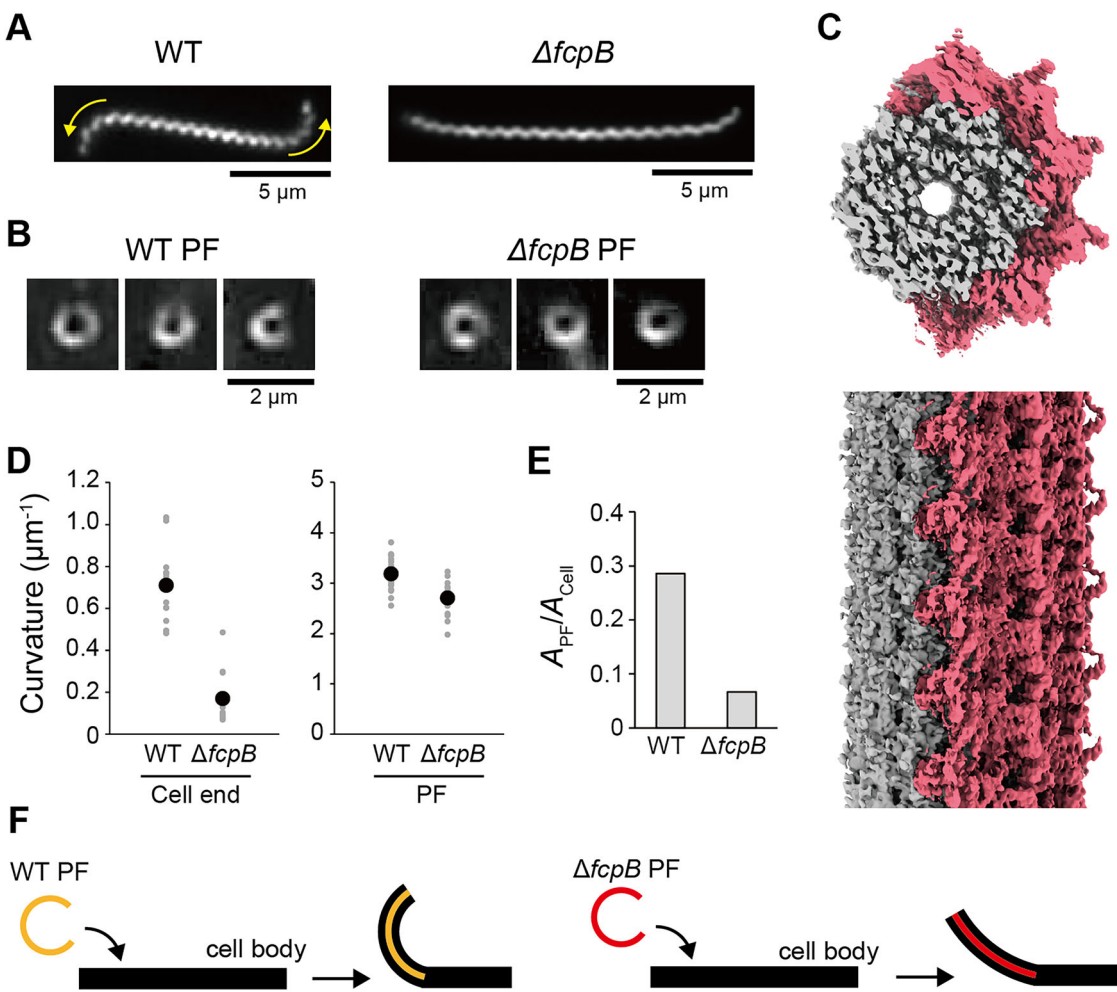

**Figure 5. FcpB increases the stiffness of PFs.**

(A) Representative morphology of WT and Δ*fcpB* (Δ*fcpB*_CL13) cells. Yellow arrows indicate curvature at the cell ends. (B) Dark-field microscopy images of purified PFs from WT and Δ*fcpB*_CL13. (C) Cryo-EM reconstruction of the PF from Δ*fcpB*_CL13, showing asymmetric localization of FcpA (magenta) on the FlaB core (gray). See Methods for details of reconstruction reproducibility. (D) Quantification of curvature at the cell ends and of isolated PFs from WT and Δ*fcpB*_CL13, based on data shown in (A, B). All individual data points are plotted, with mean values indicated by black circles. Data were collected from 16 cell ends and 20 PFs for WT, and 12 cell ends and 20 PFs for Δ*fcpB*_CL13, obtained from two independent biological replicates. Statistical significance was assessed using Student's *t*-test (cell end $P = 2.21 \times 10^{-10}$, PF $P = 2.36 \times 10^{-5}$). (E) Relative stiffness of PFs compared with the cell body, estimated from the averaged curvature values in D using linear elasticity theory. (F) Schematic illustration showing how PF stiffness affects cell-end bending in WT and Δ*fcpB* strains. In WT cells, PF stiffness is sufficient to bend the cell ends, while in the Δ*fcpB* mutant, reduced PF stiffness results in weak or absent bending. Source data are available online for this figure.

plays a critical role in mediating its interaction with sheath proteins and is essential for proper PF assembly.

## Discussion

PFs in spirochetes are composed of multiple proteins, and in some species, they consist of a core filament and an outer sheath. In *B. hyodysenteriae*, the core filament and sheath are formed by three FlaB proteins (FlaB1–3) and one FlaA protein, respectively (Li et al, 2008). Although PFs in *B. hyodysenteriae* normally exhibit a helical structure, mutants lacking any FlaB exhibit a straight sheath tube, while the FlaA-deficient mutant forms a straight core filament (Li et al, 2000, 2008). These findings suggest that the interaction between the sheath and the core filament determines

the overall morphology of the PF in this species. In this study, we investigated the PFs of *L. biflexa*, which exhibit a distinctively curved morphology, to determine the distinct roles of core and sheath proteins in PF assembly. Although the structural basis of *Leptospira* PFs has been previously explored using cryo-EM analysis of mutants from different species (Brady et al, 2022), we focused on mutants of each sheath protein within the same strain of *L. biflexa*, aiming to provide an integrated understanding of the flagellar assembly mechanism. Our ~3 Å resolution cryo-EM structures of mutant PFs lacking individual sheath proteins, as well as from a complemented strain for the sheath protein FlaA2, revealed the molecular interactions among flagellar components and clarified the mechanism that generates PF curvature. In combination with optical microscopy, we further demonstrated a novel role of the flagellar sheath in providing appropriate stiffness

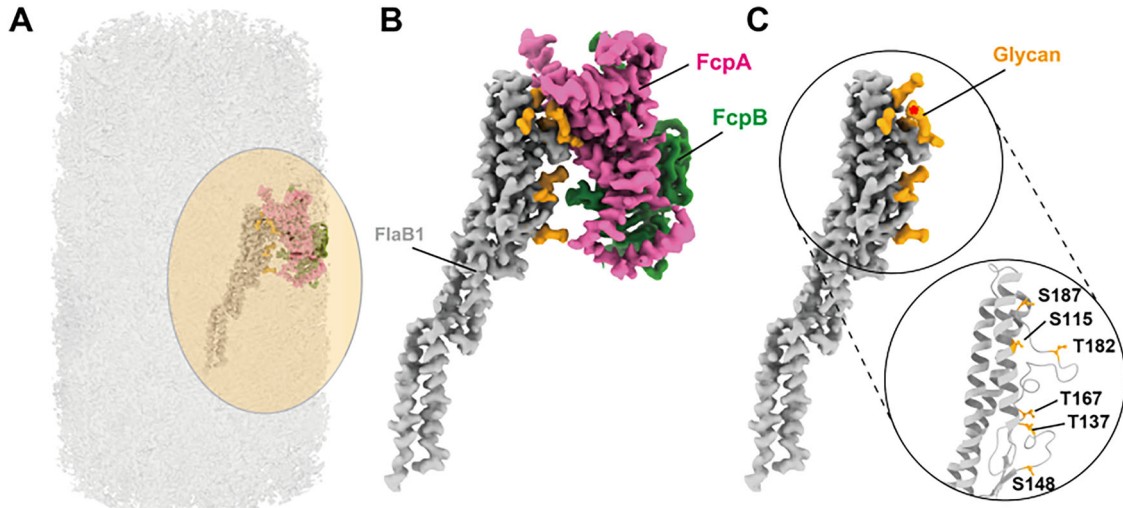

**Figure 6.    Interaction of glycosylated FlaB1 with sheath proteins FcpA and FcpB.**

(**A**) Overview of a periplasmic flagellum uniformly covered with sheath proteins FcpA (magenta) and FcpB (green). (**B**) Enlarged view of a single glycosylated FlaB1 filament (gray) associated with FcpA and FcpB. (**C**) Enlarged view of a single glycosylated FlaB1 filament. Segmented cryo-EM densities corresponding to putative glycans on serine and threonine residues of FlaB1 are shown in orange. Glycosylated residues are labeled in the close-up view, and a density of unknown corresponding residue is indicated by a red asterisk. See Methods for details of reconstruction reproducibility.

to enable periodic transformation of the cell body during locomotion.

## Potentially distinct roles of FlaB1 and FlaB2

*L. biflexa* contains four *flaB* genes (*flaB1–4*), but our previous immunoblot analysis revealed that only FlaB1 and FlaB2 were expressed and incorporated into the PFs of the WT strain (Sasaki et al, 2018). In the present study, although we could not identify the specific FlaB isoform in WT PFs due to their coiled morphology (Fig. 1), the cryo-EM reconstructions of straight PFs from mutants lacking FlaA2 and/or FcpB clearly showed that the core filament, which is uniformly covered by the sheath proteins, is composed solely of FlaB1 (Fig. 6; Appendix Figs. S6 and S12). Moreover, FlaB1 appeared to be glycosylated at three serine and three threonine residues, with $Thr_{137}$ and $Thr_{182}$ being unique to FlaB1 and absent in FlaB2 (Appendix Fig. S13). Glycosylation of flagellar filaments is known in various bacteria, including spirochetes (Li et al, 1993; Brimer and Montie, 1998; Wyss, 1998; Thibault et al, 2001), and it plays a crucial role in flagellar assembly and motility in *T. denticola* (Kurniyati et al, 2017). These glycosylated residues lie at the interface between FlaB1 and the sheath proteins (Fig. 6), which may help bridge the structural gap and suggest that the sheath attaches to the core via these glycans. Specifically, FlaB1-specific $Thr_{137}$ likely interacts with FcpB, whereas $Thr_{182}$ likely interacts with FcpA (Fig. 6). This interaction may explain why core filaments associated with sheath proteins are composed of FlaB1 rather than FlaB2. Consistent with this, the apparent molecular weight of FlaB1 on SDS-PAGE (~ 35 kDa) is higher than the theoretical weight (~31.3 kDa), supporting the presence of glycosylation. Additionally, our previous study confirmed the in vivo interaction between FcpA and FlaB1 (Sasaki et al, 2018).

Future studies employing site-directed mutagenesis of $Thr_{137}$ and $Thr_{182}$ in FlaB1 will be valuable in assessing the functional significance of these glycosylated residues in core–sheath interactions.

On the other hand, in the same preparations, the core filament lacking FcpA sheath was composed solely of FlaB2 (Appendix Figs. S6 and S12). This observation is consistent with our previous results showing that only FlaB2 was detected in the isolated *ΔfcpA* PFs, which lacked any sheath structures (Sasaki et al, 2018). Although FlaB1 was expressed in the *ΔfcpA* strain, it was not detected in the purified PFs, suggesting that FcpA plays a role in stabilizing FlaB1-based filaments. These findings imply that while FlaB2 is able to form filaments in the absence of FcpA, such filaments may be structurally unstable and subject to degradation. Notably, in our earlier work, we frequently observed that the proximal part of the PF—closer to the basal motor—was sheathed, whereas the distal part lacked sheath coverage (Sasaki et al, 2018). A similar organization has been described in *Shewanella putrefaciens*, which possesses two flagellin genes, *flaA* and *flaB*: FlaA forms the proximal portion of the polar flagellum, with FlaB polymerizing distally over it to complete the filament (Kühn et al, 2018). By analogy, it is plausible that in *L. biflexa*, the proximal region of the PF consists of FlaB1 covered with the sheath, while the distal region is composed of exposed FlaB2. Further studies are required to test this hypothesis.

## Asymmetric sheath localization requires moderate-affinity association of FlaA2

We showed that FlaA2, but not FlaA1, plays a critical role in controlling the asymmetric localization of FcpA and FcpB, which in turn generates the curved structure of PFs (Fig. 2). Although no

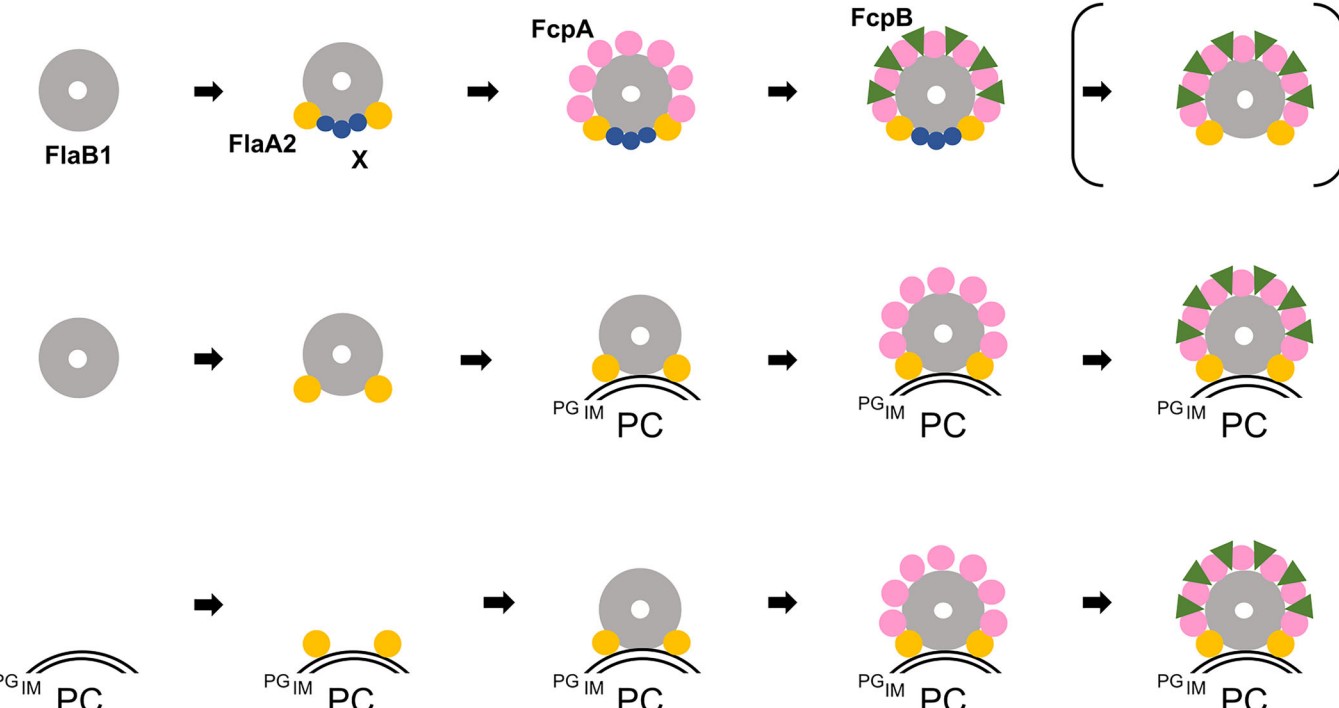

**Figure 7. Proposed models of PF assembly in *L. biflexa*.**

Top: FlaA2 binds to two specific sites on FlaB1 core, while unidentified proteins (denoted as X) mask regions of FlaB1 that are destined to form the inner curvature. FcpA, subsequently, associates with FlaA2, followed by the binding of FcpB. The unidentified proteins may dissociate upon PF curvature. Middle: FlaA2 first binds to FlaB1, and the complex anchors to the protoplasmic cylinder (PC), where FlaA2 may interact with peptidoglycans (PG), thereby masking the inner curvature of the filament. FcpA then binds to FlaA2, followed by FcpB. Bottom: FlaA2 initially associates with the PC, forming linear arrays reminiscent of rails. FlaB1 is then recruited to this scaffold, followed by the sequential binding of FcpA and FcpB. IM inner membrane, PC protoplasmic cylinder, PG peptidoglycan.

density corresponding to FlaA2 was observed in the reconstructed WT PF structure by single-particle analysis (Fig. 1), the structure of PFs from the *flaA2*-complemented Δ*flaA2* strain revealed additional densities at both edges of the localized sheath, distinct from those of FcpA and FcpB, and likely corresponding to FlaA2 (Fig. 2E). A previous study by Brady et al reported that FlaA2, along with a potential new sheath protein FlaAP forms a linear array on the FlaB1 core filament (referred to as FlaB4 in their study) in the Δ*fcpA* strain (Brady et al, 2022). However, the core filaments in their preparation exhibited considerable heterogeneity: this particular structure was observed in fewer than 5% of the analyzed particles, and in 60–80% of cases, no additional structure was apparent on the core filament. Although it is possible that FlaA2 (and perhaps FlaAP) was removed from the PFs during our sample preparation—due to the use of density gradient ultracentrifugation, which was not employed in our previous study—the extremely low frequency of FlaA2-associated PFs in our data suggests that FlaA2 binding to PFs may be inherently weak or transient. Nevertheless, because FlaA2 has been detected in PFs (Sasaki et al, 2018) and is essential for asymmetric sheath localization (Fig. 2D,E), it is plausible that FlaA2 binding to the FlaB1 core filament with moderate affinity could transiently mask a portion of the core filament and facilitate FcpA binding to one side of the core filament (discussed in more detail below using Fig. 6).

This study also demonstrated that complementation with *flaA2* alone restored the phenotypic and functional defects in the FlaA1/ FlaA2-deficient mutant, whereas the FlaA1-deficient mutant exhibited colony sizes comparable to the WT strain (Fig. 2; Appendix Fig. S4). FlaA1 is predicted to contain a signal peptide, similar to FlaA2, suggesting secretion via the Sec/SPI pathway. Both proteins possess the FlaA domain, and their predicted tertiary structures are highly similar, as confirmed by pairwise structural alignment (TM scores: 0.52 overall; 0.58 for the FlaA domain). However, multimer predictions using AlphaFold 3 (Abramson et al, 2024) and AlphaFold-Multimer (Evans et al, 2022) yielded ipTM scores of 0.37 and 0.57, respectively, which do not support a stable interaction. Thus, while structurally similar, FlaA1 and FlaA2 are unlikely to form a heteromeric complex. As with FlaA2, FlaA1 may also dissociate from PFs during purification. Future work will require targeted cryo-EM analyses and complementary labeling approaches to directly visualize the stability and spatial distribution of both FlaA2 and FlaA1 along the PF.

## FcpA is the major coiling protein

This study demonstrates that FcpA can localize to one side of the FlaB1 core filament and produce curvature of the PFs even in the absence of FcpB (Fig. 5B,C), indicating that FcpA alone plays the role of a coiling protein. The present and previous studies

demonstrated that *fcpA* knockout mutants lacked FcpB in PFs (Wunder et al, 2018). The present and previous cryo-electron microscopic and the present biochemical analyses confirmed the interaction between FcpA and FcpB (Brady et al, 2022), suggesting that FcpA acts as a platform for FcpB to dock to the sheath. In addition, FcpA may function to stabilize the PF itself, as discussed above. FlaB2 can polymerize in the absence of FcpA. In *Salmonella* flagellin, approximately 50 residues each at the N- and C-termini, termed domain D0, are responsible for the polymerization into the filament (Samatey et al, 2001). Since the domain D0 is conserved in FlaB1 and FlaB2 of *L. biflexa*, it is most likely that FlaB1 can also polymerize in the absence of FcpA. On the other hand, the fact that the core of the PF to which FcpA binds was composed of FlaB1 alone and that FlaB1 was not detected in isolated flagella of the *fcpA* knockout strain suggests that FcpA plays an important role in the stability of polymerized FlaB1.

## FcpB acts as a "wedge" reinforcing PF

Although FcpB was first identified as a flagellar coiling protein (Wunder et al, 2018), this study shows that the primary role of FcpB is to increase the flagellar stiffness. Docking of FcpB into FcpA rows does not appear to alter the structure of the rows (Fig. 3A,B), while the docking appears to stabilize the structure of FcpA, as the structure of the N-terminus of FcpA can only be determined with FcpB (Fig. 3C). Additionally, in the WT PFs, the FcpB docking would inhibit the outward bending of a curved PF due to the steric hindrance created between the FcpA and the FcpB rows. Thus, FcpB may act as a wedge, providing the rigidity necessary for the flagellum to exert a bending force on the cell body. In low-viscosity environments, thrust for swimming is generated by a combination of protoplasmic cylinder rotation and cell-end gyration (Goldstein and Charon, 1990). Since both the shape and the gyration of the cell ends are characterized by PFs, the $\Delta fcpB$-induced reduction in PF stiffness could compromise swimming motility under such conditions (Fig. 4C).

Notably, flagellar filaments form superhelical structures not only in bacteria but also in archaea (Kreutzberger et al, 2022). In spirochetal *Leptospira*, we observed a superhelical configuration in unsheathed core filaments (Fig. EV5; Appendix Fig. S14). In contrast, the uniformly sheathed PFs from both $\Delta flaA2$ and $\Delta fcpB$ mutants exhibited markedly smaller curvature than the core filaments (Fig. EV5; Appendix Fig. S14). Because FcpA contributes to coiling formation while FcpB provides rigidity, the uniform sheath likely suppressed the intrinsic superhelical curvature of the core filaments. These findings suggest that the asymmetric localization of the sheath in the wildtype maintains the superhelical shape of the core filaments, which is essential for the generation of functional PFs.

## PF assembly model

We propose models of PF assembly in *L. biflexa* that account for curvature generation and structural reinforcement (Fig. 7). In the first (top) model, FlaA2 binds at two specific positions on the FlaB1 core filament. In addition, unidentified proteins (designated here as X) may mask the portion of the filament that ultimately forms the inner curvature of the coiled structure. However, the mechanism by which FlaA2 and X recognize specific binding sites on FlaB1 remains unclear. Brady et al, (2022) suggested that longitudinal sliding between protofilaments #4 and #5 (D1 domains) creates a unique binding interface for FlaA2. They emphasized the role of glycosylated $Thr_{137}$ specific for FlaB4—which corresponds to FlaB1 in our study—for FlaA2 binding. In contrast, our cryo-EM data (Fig. 6) indicate that $Thr_{137}$, possibly modified with glycan, is exposed on all FlaB1 protofilaments and interacts directly with FcpB, suggesting that this residue is not exclusively involved in FlaA2 binding. As shown in our present analysis (Fig. 6) and previous biochemical work (Sasaki et al, 2018), FcpA interacts with both FlaA2 and FlaB1 and subsequently serves as a platform for FcpB docking (Figs. 2 and 3). It is possible that protein X dissociates from the filament after curvature is established.

The second (middle) model is based on the observation that PFs wrap tightly around the protoplasmic cylinder (PC) (Raddi et al, 2012; Takabe et al, 2017). In this scenario, FlaA2 interacts with FlaB1 and anchors the filament to the PC—possibly through recognition of peptidoglycans—thereby locally masking one side of the filament and directing asymmetric sheath assembly.

The third (bottom) model proposes that FlaA2 first associates with the PC and forms "rails" that guide subsequent FlaB1 polymerization along a specific trajectory. This could provide spatial cues for curvature of the growing filament. However, FlaA2 lacks both a conserved peptidoglycan-binding domain—present in OmpA-related outer membrane proteins and MotB—and the known peptidoglycan-association motif ($NX_2LSX_2RAX_2VX_3L$) (De Mot & Vanderleyden, 1994; Koebnik, 1995). The mechanism by which FlaA2 associates with peptidoglycan therefore remains to be elucidated.

In any of these models, our single-particle analysis suggests that the interaction between sheath proteins and the core filament—likely mediated by glycan moieties—is relatively weak. While the sheath is essential for maintaining PF shape and rigidity, it may not rotate rigidly with the FlaB1 core. Rather, as shown previously (Takabe et al, 2017), the sheath may be more closely associated with the outer membrane, and the core filament may rotate underneath it, creating a unique form of motility.

In conclusion, this study reveals the distinct roles and cooperative interactions of sheath proteins in the assembly and mechanical function of *Leptospira* PFs. Asymmetric localization of FcpA and FcpB, which is governed by FlaA2 but not FlaA1, is essential for flagellar curvature and motility. FcpA alone is sufficient to induce filament curvature, while FcpB enhances stiffness, allowing the flagellum to transmit force and deform the cell body for efficient swimming in liquid environments. Furthermore, our findings indicate that FlaB1, but not FlaB2, forms the core filaments, and that specific glycosylation of FlaB1 is required for sheath interaction. However, several key questions remain. It is still unclear how FlaA2 recognizes and localizes to the core filaments, whether an additional factor—referred to as protein X in Fig. 7—is required for this localization, and whether FlaA2 directly interacts with the PC. Moreover, whether the core filament consists exclusively of FlaB1 and what precise role FlaA2 plays in PF formation are yet to be determined. Addressing these questions will not only deepen our understanding of flagellar assembly in *Leptospira*, but also shed light on the evolutionary diversity of bacterial flagella and motility mechanisms.

# Methods

### Reagents and tools table

| Reagent/resource | Reference or source | Identifier or catalog number |
| --- | --- | --- |
| **Experimental models** | | |
| *Leptospira biflexa* serovar Patoc strain Patoc I 42L2 | Sasaki et al, 2018 | N/A |
| *Escherichia coli* strain π1 | Demarre et al, 2005 | N/A |
| *Escherichia coli* strain β2163 | Demarre et al, 2005 | N/A |
| **Recombinant DNA** | | |
| pSW29T | Demarre et al, 2005 | N/A |
| pAT21 | Trieu-Cuot and Courvalin, 1983 | N/A |
| pCj94SpLe94 | Picardeau, 2008 | N/A |
| pNKLbKmR | This study | N/A |
| **Antibodies** | | |
| Rabbit anti-FcpA | Sasaki et al, 2018 | N/A |
| Rabbit anti-FcpB | This study | N/A |
| Rabbit anti-FlaA1 | This study | N/A |
| Rabbit anti-FlaA2 | Sasaki et al, 2018 | N/A |
| Goat anti-rabbit IgG HRP | Bio-Rad | 1706515 |
| **Oligonucleotides and other sequence-based reagents** | | |
| PCR primers | This study | Appendix Table S2 |
| **Chemicals, enzymes and other reagents** | | |
| Leptospira medium base EMJH | BD | 279410 |
| Bovine serum albumin (EMJH medium) | Millipore | 81-003-3 |
| Zinc sulfate (EMJH medium) | FUJIFILM Wako Pure Chemical | 268-00422 |
| Magnesium chloride (EMJH medium) | FUJIFILM Wako Pure Chemical | 136-03995 |
| Calcium chloride (EMJH medium) | FUJIFILM Wako Pure Chemical | 033-25035 |
| Iron(II) Sulfate (EMJH medium) | FUJIFILM Wako Pure Chemical | 094-01082 |
| Sodium pyruvate (EMJH medium) | FUJIFILM Wako Pure Chemical | 199-03062 |
| Glycerol (EMJH medium) | FUJIFILM Wako Pure Chemical | 072-04945 |
| Tween 80 (EMJH medium) | Kanto Chemical | 40353-32 |
| Cyanocobalamin (EMJH medium) | FUJIFILM Wako Pure Chemical | 224-00344 |
| LB medium | Kanto Chemical | 24308-08 |
| Agar | BD | 214010 |
| Thymidine | FUJIFILM Wako Pure Chemical | 207-19421 |
| 2,6-Diaminopimelic acid | FUJIFILM Wako Pure Chemical | 044-18471 |
| Spectinomycin | FUJIFILM Wako Pure Chemical | 113-00343 |

| Reagent/resource | Reference or source | Identifier or catalog number |
| --- | --- | --- |
| Kanamycin | FUJIFILM Wako Pure Chemical | 115-00342 |
| DNeasy Blood & Tissue Kit | Qiagen | 69506 |
| DNA polymerase | TaKaRa | R050A |
| BigDye Terminator v3.1 Cycle Sequencing Kit | Applied Biosystems | 4337455 |
| NEBuilder HiFi DNA Assembly | New England Biolabs | E2621L |
| *Sal*I | TaKaRa | 1080 A |
| Triton X-100 | Millipore | 648466 |
| Protein G agarose | Thermo Scientific | 20398 |
| QIAseq FX DNA Library kit | Qiagen | 180477 |
| Methylcellulose | Sigma-Aldrich | M0262 |
| Sucrose | FUJIFILM Wako Pure Chemical | 196-00015 |
| EDTA·2Na | FUJIFILM Wako Pure Chemical | 345-01865 |
| Sodium Hydroxide | FUJIFILM Wako Pure Chemical | 192-15985 |
| Sodium chloride | FUJIFILM Wako Pure Chemical | 191-01665 |
| **Software** | | |
| ImageJ | https://imagej.net/ij/index.html | N/A |
| Sickle | Joshi and Fass, 2011 | N/A |
| BWA | Li and Durbin, 2010 | N/A |
| SAMtools | Li, 2011 | N/A |
| VarScan | Koboldt et al, 2012 | N/A |
| AlphaFold 3 | Abramson et al, 2024 | N/A |
| AlphaFold-Multimer | Evans et al, 2022 | N/A |
| EPU software | Thermo Fisher Scientific | N/A |
| SerialEM | Mastronarde, 2005 | N/A |
| MotionCorr2 | Zheng et al, 2017 | N/A |
| Gctf | Zhang, 2016 | N/A |
| RELION 3.1 | Zivanov et al, 2018 | N/A |
| cryoSPARC v4.2.1 | Punjani et al, 2017 | N/A |
| ModelAngelo | Jamali et al, 2024 | N/A |
| UCSF ChimeraX | Goddard et al, 2018 | N/A |
| COOT | Emsley et al, 2010 | N/A |
| Phenix | Liebschner et al, 2019 | N/A |
| **Other** | | |
| Glass beads | Nippon Gene | 314-06251 |
| 1-mm glass beads | Sigma-Aldrich | Z250473 |
| 3500 Genetic Analyzer | Applied Biosystems | |
| MiSeq | Illumina | |

| Reagent/resource | Reference or source | Identifier or catalog number |
|---|---|---|
| Dark-field microscope | Olympus | BX53,UPlan-FLN 100×, U-DCW |
| CMOS camera | Photron | IDP |
| CCD camera | Watec | WAT-910HX/RC |
| swing rotor SW 32 Ti | BECKMAN COULTER | 369650 |
| Gradient Fractionator | BIOCOMP | 153 |
| Rotor 50.2 Ti | BECKMAN COULTER | 337901 |
| Quantifoil holey carbon grid (R1.2/1.3 Cu 200mech) | Quantifoil | N1-C14nCu20-01 |
| Vitrobot Mark IV | Thermo Fisher Scientific | |
| Titan Krios | Thermo Fisher Scientific | |
| Cs corrector | CEOS | |
| Falcon3EC CMOS direct electron detector | Thermo Fisher Scientific | |
| K3 direct electron detector | GATAN | |

## Bacteria and media

*L. biflexa* serovar Patoc strain Patoc I (42L2) was cultured in liquid Ellinghausen–McCullough–Johnson–Harris (EMJH) medium at 30 °C (Faine et al, 1999). Kanamycin and/or spectinomycin were added at a final concentration of 25 μg/ml when required. *E. coli* strains π1, used for DNA cloning, and strain β2163, used as the donor for conjugation, were grown in LB medium supplemented with 0.3 mM thymidine (for π1) or 0.3 mM diaminopimelate (for β2163) (Demarre et al, 2005).

## *Leptospira* growth, colony formation, and soft-agar motility assays

Mid-log phase cultures of *L. biflexa* strains (optical density at 420 nm [$OD_{420}$] = 0.3–0.4) were adjusted to an $OD_{420}$ of 0.3, and 2 μL of the culture was inoculated into 2 mL of liquid EMJH medium. $OD_{420}$ was measured at 0, 24, 33, 48, 51, 72, 81, 96, and 105 h after inoculation; measurements were terminated at 105 h (day 4), as sedimentation occurred after day 5. Each inoculation was performed in triplicate, and the experiment was repeated three times.

For colony formation, cultures adjusted to an $OD_{420}$ of 0.3 were further diluted $10^6$-fold in liquid EMJH medium, and 200 μL of the diluted suspension was spread onto EMJH soft agar (0.4%) plates using glass beads (Bac'n'Roll Beads; Nippon Gene, Tokyo, Japan). Plates were incubated at 30 °C for 7 days. The assay was performed independently twice.

For soft-agar motility assay, a 200-μL pipette tip was cut at the 20-μL mark, and the modified tip was used to punch a 10-μL well into EMJH soft agar (0.4%) plates. Into each well, 2 μL of culture adjusted to an $OD_{420}$ of 0.3 was inoculated. Plates were incubated at 30 °C for 5 days. The assay was performed twice independently, using four plates in each experiment.

Colony images were acquired with a Limited STAGE II transilluminator (AMZ System Science) equipped with a Power-Shot G7X Mark II digital camera (Canon). Colony areas were quantified in ImageJ by applying a manually set brightness threshold, with regions above the threshold defined as colony areas. Colonies that overlapped were excluded from the analysis.

## Construction of gene knockout mutants and complemented strains

To construct *fcpB*, *flaA1*, and *flaA2* knockout mutants, upstream and downstream regions of these genes were amplified from genomic DNA of *L. biflexa* 42L2 strain. The spectinomycin resistance cassette was amplified as previously described (Sasaki et al, 2018). PCR products were cloned into *Cla*I-digested pSW29T (Demarre et al, 2005) using NEBuilder HiFi DNA Assembly (New England Biolabs). The resulting plasmid was transformed into *E. coli* β2163 and then introduced into *L. biflexa* by conjugation as previously described (Slamti and Picardeau, 2012). After conjugation, cells were spread onto EMJH soft agar (0.4% agar) plates containing spectinomycin using the glass beads, and then incubated at 30 °C. Spectinomycin-resistant transformants were screened for allelic exchange in the target genes by PCR and immunoblotting.

For complementation of spectinomycin-resistant knockout mutants, the kanamycin resistance cassette was amplified from pAT21 (Trieu-Cuot and Courvalin, 1983) and cloned into cloned into the PCR-generated pCjSpLe94 (Picardeau, 2008) using NEBuilder HiFi DNA Assembly (New England BioLabs), generating pNKLbKmR. The *flaA2* gene including its putative promoter region, and the *flaA1/flaA2* operon were amplified from genomic DNA of *L. biflexa* strain Patoc I (42L2) and cloned into the *Sal*I-digested pNKLbKmR. These plasmids were introduced into the *L. biflexa* Δ*flaA2* mutant as described above. Primers used in this study are listed in Appendix Table S2.

## Isolation and purification of PFs

PFs were purified using a modified method based on Li et al, (2000). *Leptospira* cells (250 mL culture, ~$7 \times 10^{10}$ cells) were harvested by centrifugation at 8,000×g for 20 min at 4 °C. The pellet was resuspended and washed twice with 50 mL of cold PBS. To remove the outer membrane sheath, the cells were resuspended in 22.5 mL of 1 M Tris-HCl (pH 7.5), and 2.5 mL of 10% Triton X-100 (final concentration: 1%) was added. The mixture was incubated for 1 h at 37 °C. The lysate was centrifuged at 4000×g for 30 min at 4 °C, and the pellet was resuspended in 2.5 mL of PBS. To shear PFs from the cell cylinders, the suspension was vortexed for 1 min with 1-mm glass beads (one-tenth volume), and the supernatant was collected by centrifugation at 18,000×g for 30 min at 4 °C. PFs were pelleted from the supernatant by centrifugation at 100,000×g for 30 min at 4 °C and resuspended in 150 μL of water. This method yielded PF fraction with fewer contaminants compared to a previously used method (Wunder et al, 2016) (Appendix Fig. S7). For cryo-EM analysis, PFs were further purified using a 20–50% (w/w) continuous sucrose density gradient in buffer C (10 mM Tris-HCl pH 8.0, 5 mM EDTA-NaOH, 1% Triton X-100), centrifuged in a swing rotor at 49,100×g for 18 h at 4 °C. Fractions (700 μL each) were collected using a gradient fractionator (BIOCOMP, NB, Canada). Peak fractions were pooled and

concentrated by ultracentrifugation at 90,000×g for 60 min at 4 °C. The pellet was resuspended in 25 μL of Buffer S (25 mM Tris-HCl pH 8.0, 1 mM EDTA-NaOH, 50 mM NaCl).

## SDS-PAGE and immunoblotting

Protein samples (1.5 μg of purified PFs without sucrose density gradient ultracentrifugation or lysates from $1.5 \times 10^8$ leptospiral cells) were prepared in SDS-PAGE sample buffer and separated on 10% SDS-PAGE gels, followed by Western blotting. Blots were probed with antisera raised against peptide fragments of FcpB (NH2-SNKHYKKNVEALKG-COOH) or FlaA1 (NH2-CGGSNGSSQNK-COOH), or with antisera against FcpA and FlaA2 (Sasaki et al, 2018).

## Immunoprecipitation

Immunoprecipitation of whole-cell lysates from *L. biflexa* WT strain with anti-FcpA antiserum was performed as previously described (Sasaki et al, 2018).

## MALDI-TOF-MS/MS

Purified PFs from *L biflexa* WT strain were subjected to 10% SDS-PAGE, and the upper protein band indicated by the asterisk in Appendix Fig. S1 was analyzed by tandem mass spectrometer, as described previously (Kawakita et al, 2016).

## Whole genome sequencing

Genomic DNA was extracted from the *ΔfcpB*_CL15 and its parent strain using DNeasy Blood & Tissue Kits (Qiagen). DNA libraries for MiSeq (Illumina) were prepared using the QIAseq FX DNA Library Kit (Qiagen) according to the manufacturer's instructions and sequenced on a MiSeq (Illumina) as $2 \times 300$ bp paired-end reads. Short reads were trimmed using Sickle (Joshi and Fass, 2011) with default parameters, followed by mapping to the *L. biflexa* serovar Patoc strain Patoc I (Ames) genomes (GenBank accession no. CP000777, CP000778, and CP000779) by using BWA v.0.7.17 (Li & Durbin, 2010). Single nucleotide variants (SNVs) and insertions/deletions (indels) in the core genome were identified using SAMtools v.1.7 (Li, 2011) and VarScan v.2.4.3 (Koboldt et al, 2012) with default parameters. The unique genomic variants to *ΔfcpB*_CL15 were extracted manually.

## Cryo-EM data acquisition and helical reconstruction

2.6 μL of the PF solution (purified as described above) were applied to a glow-discharged Quantifoil holey carbon grid (R1.2/1.3, Cu, 200 mesh), blotted for 4 s at 4 °C, and plunge-frozen in liquid ethane using a Vitrobot Mark IV (Thermo Fisher Scientific). The grid was inserted into a Titan Krios (Thermo Fisher Scientific), which was operated at an acceleration voltage of 300 kV and equipped with a Cs corrector (CEOS, GmbH). Images were acquired using a Falcon3EC CMOS direct electron detector (Thermo Fisher Scientific) in counting mode. For the *ΔflaA2* mutant PFs, data were automatically collected using EPU software (Thermo Fisher Scientific) at a physical pixel size of 1.13 Å, with 109 frames at a dose of 0.46 e⁻/Å² per frame, an exposure time of

84.18 s per movie, and defocus ranging from −0.5 to −2.0 μm. A total of 2318 movies were collected. For the *ΔfcpB* mutant PFs, data were automatically collected using EPU software at a physical pixel size of 1.13 Å, with 108 frames at a dose of 0.46 e⁻/Å² per frame, an exposure time of 83.81 sec per movie, and defocus ranging from −0.5 to −2.0 μm. A total of 1645 movies were collected.

For the *ΔflaA2* mutant PFs, the movie frames were subjected to beam-induced motion correction using MotionCorr2.1 (Zheng et al, 2017), and the contrast transfer function (CTF) was assessed using Gctf (Zhang, 2016). Motion correction and CTF estimation were performed using RELION 3.1 (Zivanov et al, 2018). Approximately 4000 particles were manually selected from 15 micrographs to perform two-dimensional (2D) classification. Using a high-quality 2D class average image as a template, a total of 1,953,561 particle images were automatically selected from all micrographs in RELION 3.1 and extracted with a box size of 90 pixels and 4×binning. After two rounds of 2D classification, 1,171,169 particles (core filament) and 438,642 particles (core filament with FcpA and FcpB) were selected and subjected to 3D classification. Subsequently, 273,749 particles (core filament) and 138,670 particles (core filament with FcpA and FcpB) were re-extracted at a pixel size of 1.13 Å and processed through five rounds of helical refinement, three rounds of CTF refinement, and Bayesian polishing. The 3D refinement and post-processing yielded a map with a global resolution of 2.31 Å (core filament) and 2.32 Å (core filament with FcpA and FcpB), according to Fourier shell correlation (FSC) with the 0.143 criterion. The refined values of helical rise and twist were 4.79 Å and 65.41° (core filament) and 4.84 Å and 65.43° (core filament with FcpA and FcpB), respectively. Local resolution was estimated using RELION 3.1. The processing workflow is shown in Appendix Fig. S6.

For the *ΔfcpB*_CL15 mutant PFs, the movie frames were processed using the same workflow described above. Approximately 3,000 particles were manually selected for 2D classification, and a total of 1,171,169 particles were automatically selected and extracted as in the previous dataset. After two rounds of 2D classification, 505,991 particles (core filament) and 525,188 particles (core filament with FcpA) were selected and subjected to 3D classification. A total of 350,712 particles (core filament) and 203,738 particles (core filament with FcpA) were re-extracted and further refined using five rounds of helical refinement, three rounds of CTF refinement, and Bayesian polishing. The final maps reached global resolutions of 2.37 Å for both structures. The refined helical parameters were 4.79 Å and 65.42° (core filament) and 4.82 Å and 65.44° (core filament with FcpA). Local resolution estimation and the full processing workflow are shown in Appendix Fig. S12.

To examine the possibility of a supercoiled structure, particles were extracted using a larger box size, and structural analyses were performed without applying helical symmetry. For the *ΔflaA2* mutant PFs, a total of 2,674,014 particle images were automatically selected from all micrographs in cryoSPARC and extracted with a box size of 720 pixels. After two rounds of 2D classification, 1,285,093 particles (core filament) and 757,983 particles (core filament with FcpA and FcpB) were selected and subjected to helical refinement without applying helical parameters, followed by cryoSPARC 3D variability analysis. Subsequently, 247,498 particles (core filament) and 230,835 particles (core filament with FcpA and FcpB) were selected and processed through homogenous refinement, CTF refinement, and helical refinement without applying

helical parameters. The helical refinement without applying helical parameters yielded maps with global resolutions of 3.66 Å (core filament) and 3.35 Å (core filament with FcpA and FcpB), according to FSC with the 0.143 criterion. The processing workflow is shown in Fig. EV5.

For the *ΔfcpB*_CL15 mutant PFs, a total of 1,996,675 particle images were automatically selected from all micrographs in cryoSPARC and extracted with a box size of 720 pixels. After two rounds of 2D classification, 563,191 particles (core filament) and 988,697 particles (core filament with FcpA) were selected and subjected to helical refinement without applying helical parameters, followed by cryoSPARC 3D variability analysis. Subsequently, 207,684 particles (core filament) and 343,980 particles (core filament with FcpA) were selected and processed through homogenous refinement, CTF refinement and helical refinement without applying helical parameters. The helical refinement without applying helical parameters yielded maps with global resolutions of 3.80 Å (core filament) and 3.23 Å (core filament with FcpA), according to FSC with the 0.143 criterion. The processing workflow is shown in Appendix Fig. S14.

## Cryo-EM data acquisition and image processing of asymmetric filaments

4 μL of the PF solution were applied to a glow-discharged Quantifoil holey carbon grid and prepared as described above. For the WT PFs, images were acquired using a Falcon3EC CMOS direct electron detector in counting mode. To analyze the asymmetric structure of the PF, data were collected under two conditions: with the stage tilted 30°, and without tilting. Both datasets were acquired automatically using EPU software at a physical pixel size of 1.13 Å, with 109 frames at a dose of 0.46 e-/Å2 per frame, an exposure time of 83.81 s per movie, and a defocus ranging from −0.5 to −2.0 μm. A total of 2614 movies (1046 micrographs at 30° tilt and 1568 micrographs without tilt) were collected. Motion correction and CTF estimation were performed using cryoSPARC v4.2.1(Punjani et al, 2017). Approximately 400 particles were manually selected from 12 micrographs to perform 2D classification. Using a high-quality 2D class average image as a template, a total of 855,996 particle images were automatically selected from all micrographs in cryoSPARC and extracted with a box size of 128 pixels and 4× binning. After two rounds of 2D classification, 77,046 particles (core filament) and 598,513 particles (sheathed filament) were selected and subjected to ab-initio reconstruction. Subsequently, 77,007 particles (core filament) and 457,393 particles (sheathed filament) were re-extracted at a pixel size of 1.13 Å and processed through two rounds of helical refinement (without applying helical parameters), followed by homogeneous refinement, non-uniform refinement, CTF refinement, and local motion correction. The helical refinement yielded maps at global resolutions of 4.35 Å (core filament) and 3.24 Å (sheathed filament), according to FSC with the 0.143 criterion. Local resolution was estimated using cryoSPARC v4.2.1 (Punjani et al, 2017). The processing workflow is shown in Fig. EV1.

For the *flaA2*-complemented *ΔflaA2* mutant PFs, images were acquired using a K3 direct electron detector (Gatan) in CDS mode. To analyze the asymmetric PF structures, data was collected under two conditions as described above. Both datasets were acquired automatically using SrialEM (Mastronarde, 2005) at a physical pixel

size of 0.87 Å, with 40 frames at a dose of 1 e-/Å2 per frame, an exposure time of 3.782 s per movie, and defocus ranging from −0.5 to −1.0 μm. A total of 6615 movies (7545 micrographs at 30° tilt and 6615 micrographs without tilt) were collected. Approximately 600 particles were manually selected from 16 micrographs. Based on 2D classification templates, 5,303,880 particles images were automatically selected and extracted. After two rounds of 2D classification, 785,401 particles (core filament) and 4,010,231 particles (sheathed filament) were selected. Subsequently, 457,214 particles (core filament) and 2,667,344 particles (sheathed filament) were re-extracted at a pixel size of 1.16 Å and subjected to the same processing steps as above. The final maps yielded global resolutions of 3.32 Å (core filament) and 2.83 Å (sheathed filament). The processing workflow is shown in Appendix Fig. S8.

For the *ΔfcpB*_CL13 mutant PFs, images were also acquired using a K3 direct electron detector in CDS mode and SrialEM, with the same acquisition settings as above. A total of 6615 movies (10,179 micrographs at 30° tilt and 7006 micrographs without tilt) were collected. Approximately 700 particles were manually selected from 10 micrographs, and 8,206,013 particle images were automatically selected and extracted. After two rounds of 2D classification, 1,436,926 particles (core filament) and 6,206,726 particles (sheathed filament) were selected. Of these, 832,792 particles (core filament) and 3,125,674 particles (sheathed filament) were re-extracted at a pixel size of 1.16 Å and processed similarly. The resulting maps had global resolutions of 3.05 Å (core filament) and 2.70 Å (sheathed filament). The processing workflow is shown in Appendix Fig. S11.

## Model building

Atomic models of the FlaB2 core filaments, the FlaB1 core filament with FcpA, and the FlaB1 core filament with FcpA and FcpB were initially generated using ModelAngelo (Jamali et al, 2024). These models were then manually fitted into the EM density maps using UCSF ChimeraX (Goddard et al, 2018). Each domain was manually remodeled and iteratively refined using COOT (Emsley et al, 2010) and Phenix program (Liebschner et al, 2019). All structural figures were prepared using UCSF ChimeraX. The statistics for 3D reconstruction and model refinement are summarized in Appendix Table S3.

## Measurement of swimming speeds

A suspension of *Leptospira* cells grown in EMJH medium at 30 °C for 3 days was introduced into a flow chamber and observed using a dark-field microscope (BX53, UPlan-FLN 40×, oil condenser U-DCW; Olympus). Cell motility was recorded with a high-speed CMOS video camera (IDP; Photron) at a frame rate of 60 frames per second. Swimming speeds of individual cells were measured by ImageJ with a custom Visual Basic macro. Cells that lacked self-propelled motility and exhibited only Brownian motion were excluded from the analysis.

## Dark-field observation of PFs and quantification of curvature of PFs and cell ends

To observe purified PFs by dark-field microscopy, coverslips were cleaned by soaking in 10 N KOH overnight and rinsing with distilled water. PFs were placed on the cleaned glasses and observed using a dark-field microscope (BX53, UPlan-FLN 100×, oil

condenser U-DCW; Olympus), and images were recorded using a high-sensitivity CCD camera (WAT-910HX/RC; Watec). To quantify cell-end curvature, UV light was applied to the *Leptospira* suspensions to immobilize the cells, as curvature could not be measured while they were motile. The curvatures of the cell ends and the isolated PFs were determined in ImageJ with a custom Visual Basic macro. PFs that appeared truncated (short fragments) and cells that remained motile after UV exposure were excluded from curvature analysis.

## Estimation of PF stiffness

The stiffness of PFs was estimated based on linear elasticity theory (Li et al, 2008) by determining the ratio of the bending moduli of the PF to the cell body. The bending moment $M$ required to deform an elastic material with the bending modulus $A$ is given by $A \triangle \kappa$, where $\Delta \kappa$ is the change in curvature. Since *Leptospira* cells lacking PFs are straight (i.e., $\kappa = 0$), the moment required to bend the cell end to a curvature of $\kappa_{\mathrm{Cell+PF}}$ is estimated as $M_{\mathrm{Cell}} = A_{\mathrm{Cell}} \kappa_{\mathrm{Cell+PF}}$, where $A_{\mathrm{Cell}}$ is the bending modulus of the cell body. Meanwhile, an isolated PFs with the curvature of $\kappa_{\mathrm{PF}}$ is bent to $\kappa_{\mathrm{Cell+PF}}$ upon association with the cell body, the moment for this deformation is given by $M_{\mathrm{PF}} = A_{\mathrm{PF}}(\kappa_{\mathrm{PF}} - \kappa_{\mathrm{Cell+PF}})$, where $A_{\mathrm{PF}}$ is the bending modulus of PF. At mechanical equilibrium, $M_{\mathrm{Cell}} + M_{\mathrm{PF}} = 0$, leading to the equation:

$$A_{\mathrm{Cell}} \kappa_{\mathrm{Cell+PF}} + A_{\mathrm{PF}}(\kappa_{\mathrm{Cell+PF}} - \kappa_{\mathrm{PF}}) = 0$$

Solving for the relative stiffness $A_{\mathrm{PF}}/A_{\mathrm{Cell}}$ yields:

$$\frac{A_{\mathrm{PF}}}{A_{\mathrm{Cell}}} = \frac{\kappa_{\mathrm{Cell+PF}}/\kappa_{\mathrm{PF}}}{1 - (\kappa_{\mathrm{Cell+PF}}/\kappa_{\mathrm{PF}})}$$

## Data availability

The cryo-EM density maps have been deposited in the Electron Microscopy Data Bank under the following accession codes: EMD-66649 (asymmetric sheathed filament of the wildtype; https://www.ebi.ac.uk/emdb/EMD-66649), EMD-66641 (core filament of the wildtype; https://www.ebi.ac.uk/emdb/EMD-66641), EMD-63347 (symmetric sheathed filament of the *ΔflaA2* mutant; https://www.ebi.ac.uk/emdb/EMD-63347), EMD-63348 (core filament of the *ΔflaA2* mutant; https://www.ebi.ac.uk/emdb/EMD-63348), EMD-66646 (asymmetric sheathed filament of the *flaA2*-complemented strain; https://www.ebi.ac.uk/emdb/EMD-66646), EMD-66642 (core filament of the *flaA2*-complemented strain; https://www.ebi.ac.uk/emdb/EMD-66642), EMD-66647 (asymmetric sheathed filament of the *ΔfcpB*_CL13 mutant; https://www.ebi.ac.uk/emdb/EMD-66647), EMD-66643 (core filament of the *ΔfcpB*_CL13 mutant; https://www.ebi.ac.uk/emdb/EMD-66643), EMD-63349 (symmetric sheathed filament of the *ΔfcpB*_CL15 mutant; https://www.ebi.ac.uk/emdb/EMD-63349), and EMD-63350 (core filament of the *ΔfcpB*_CL15 mutant; https://www.ebi.ac.uk/emdb/EMD-63350). Atomic coordinates have been deposited in the Protein Data Bank under accession codes 9X80 (asymmetric sheathed filament of the wildtype; https://www.rcsb.org/structure/9X80), 9X7K (core filament of the wildtype; https://www.rcsb.org/structure/9X7K), 9LRY (symmetric sheathed filament of the *ΔflaA2* mutant; https://www.rcsb.org/structure/9LRY), 9LRZ (core filament of the *ΔflaA2* mutant; https://www.rcsb.org/structure/9LRZ), 9X7S (asymmetric sheathed filament of the *flaA2*-complemented strain; https://www.rcsb.org/structure/9X7S), 9X7L (core filament of the *flaA2*-complemented strain; https://www.rcsb.org/structure/9X7L), 9X7V (asymmetric sheathed filament of the *ΔfcpB*_CL13 mutant; https://www.rcsb.org/structure/9X7V), 9X7M (core filament of the *ΔfcpB*_CL13 mutant; https://www.rcsb.org/structure/9X7M), 9LS0 (symmetric sheathed filament of the *ΔfcpB*_CL15 mutant; https://www.rcsb.org/structure/9LS0), and 9LS1 (core filament of the *ΔfcpB*_CL15 mutant; https://www.rcsb.org/structure/9LS1). Other data are available from the corresponding author upon reasonable request.

The source data of this paper are collected in the following database record: biostudies:S-SCDT-10_1038-S44318-026-00731-1.

## Peer review information

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

## Acknowledgements

We thank Aya Takamori for her kind help with mass spectrometry measurements, which were supported by the general supporting team for the Grant-in-Aid for Scientific Research on Innovative Areas "Harmonized Supramolecular Motility Machinery and Its Diversity" (25117501) directed by Makoto Miyata. We also thank Genji Kurisu, Keiichi Namba, Takayuki Kato, Kyosuke Takabe, Yuri Sano, and Yuya Sasaki for their technical advice and assistance. This work was supported by JSPS KAKENHI grant numbers 21H02727 (SN), 22K07062 (NK), and 24K02274 (SN) and JST, PRESTO grant number JPMJPR21E5 (AK).

## Author contributions

**Akihiro Kawamoto**: Conceptualization; Data curation; Formal analysis; Funding acquisition; Validation; Investigation; Visualization; Methodology; Writing—original draft; Project administration; Writing—review and editing. **Toshiki Kuribayashi**: Investigation; Writing—review and editing. **Masatomo Morita**: Investigation; Writing—review and editing. **Shuichi Nakamura**: Conceptualization; Data curation; Formal analysis; Funding acquisition; Validation; Investigation; Visualization; Methodology; Writing—original draft; Writing—review and editing. **Nobuo Koizumi**: Conceptualization; Data curation; Formal analysis; Funding acquisition; Validation; Investigation; Visualization; Methodology; Writing—original draft; Writing—review and editing.

Source data underlying figure panels in this paper may have individual authorship assigned. Where available, figure panel/source data authorship is listed in the following database record: biostudies:S-SCDT-10_1038-S44318-026-00731-1.

## Disclosure and competing interests statement

The authors declare no competing interests.

# Expanded View Figures

**Figure EV1.   Summary of Cryo-EM data acquisition and image processing of periplasmic flagella (PFs) from wild-type strain.**

(**A**) Cryo-EM image of purified PFs. Blue and red arrows indicate the unsheathed core filament and the sheathed filament, respectively. (**B**) Data-processing workflow. (**C**) Reconstructed image of the unsheathed core filament. (**D**) Reconstructed image of the sheathed filament. (**E**) 2D class image of the unsheathed core filament. (**F**) 2D class image of the sheathed filament. (**G**) Fourier shell correlation (FSC) analysis of the reconstructed unsheathed core filament, showing a global resolution of 4.35 Å. (**H**) FSC analysis of the reconstructed sheathed filament, showing a global resolution of 3.24 Å. (**I–L**) Fitted atomic models and corresponding density maps of FlaB2 (**I**), FlaB1 (**J**), FcpA (**K**), and FcpB (**L**).

▶

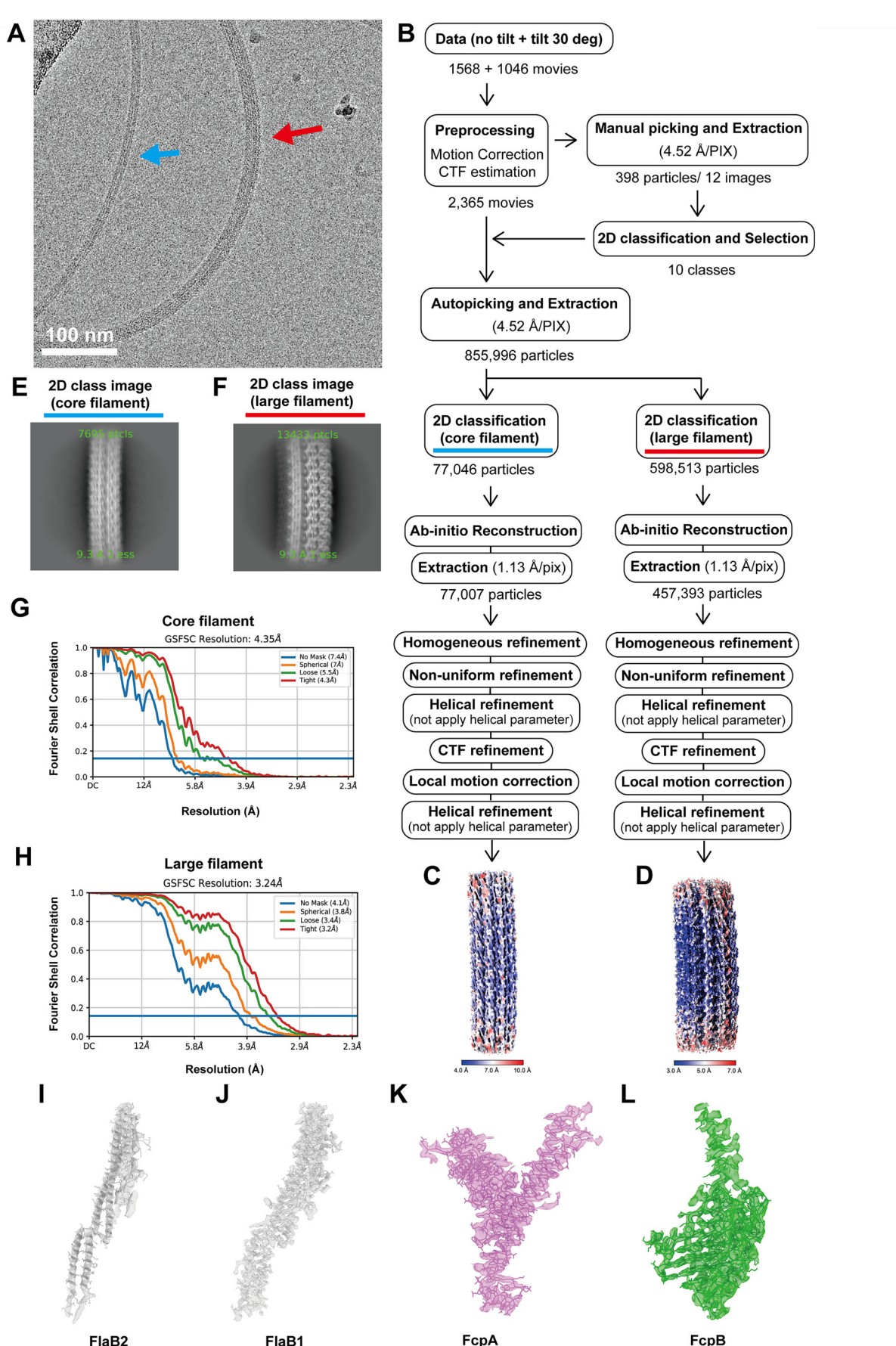

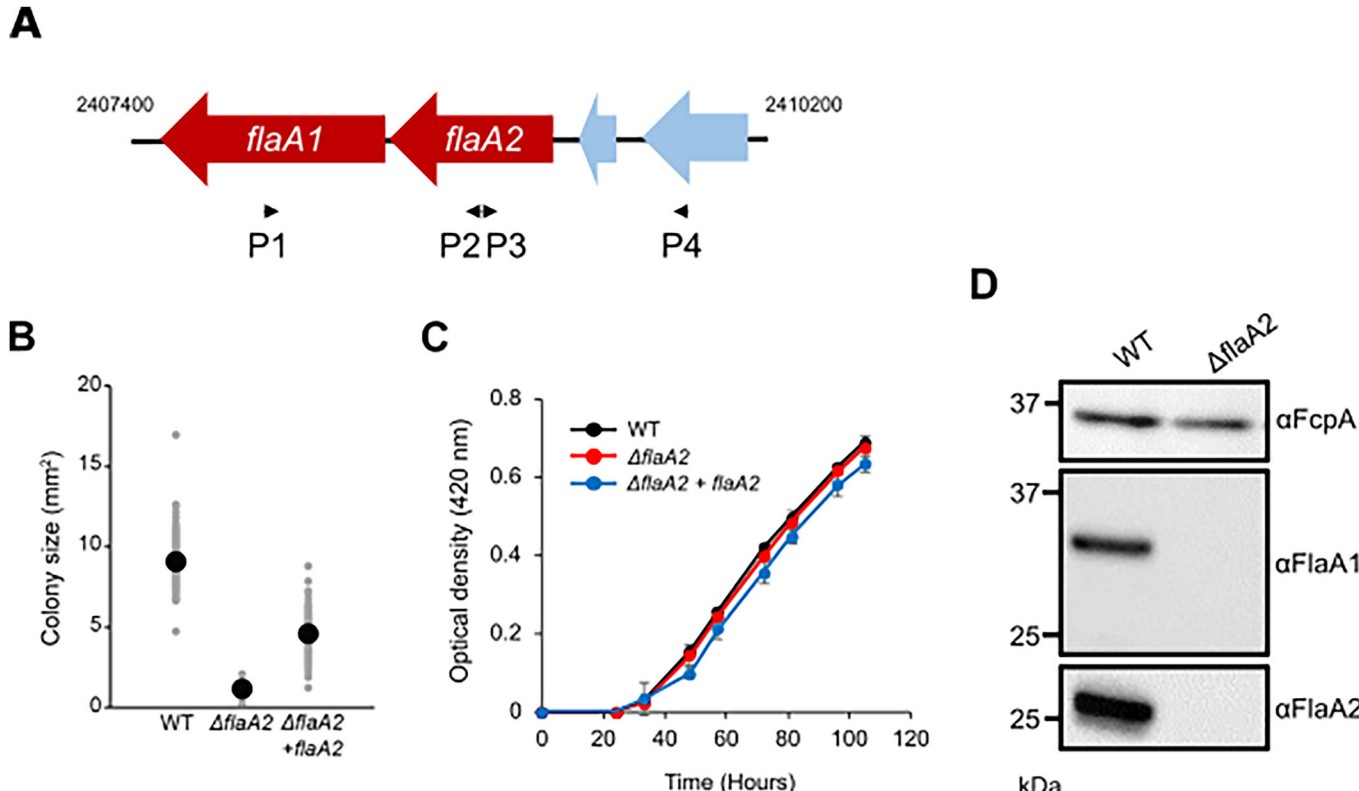

**Figure EV2. *ΔflaA2* mutant expresses neither FlaA2 nor FlaA1.**

(A) Schematic diagram of the genomic region (positions 2407400 to 2410200) of *L. biflexa* serovar Patoc strain Patoc I (Paris) (GenBank accession no. CP000786.1), showing the relative positions and orientations of *flaA2* and *flaA1*. The locations of primers used for constructing the *flaA2* deletion mutant (P1: 2407983–2408003, P2: 2408872–2408895, P3: 2408896–2408916, P4: 2409784–2409803) are indicated by black arrows. (B) Quantification of colony size of wild-type (WT), *ΔflaA2*, and *flaA2*-complemented (*ΔflaA2 + flaA2*) strains. strains. All individual data points are plotted, with mean values indicated by black circles. Data were collected from 131 colonies for WT, 194 colonies for *ΔflaA2*, and 176 colonies for *flaA2*-complemented srains, obtained from two independent biological replicates. Statistical significance was assessed using a two-tailed Student's *t*-test, assuming unequal variances (WT vs *ΔflaA2* $P = 6.94 \times 10^{-96}$, WT vs *ΔflaA2 + flaA2* $P = 8.29 \times 10^{-75}$, *ΔflaA2* vs *ΔflaA2 + flaA2* $P = 2.50 \times 10^{-94}$). (C) Growth of *L. biflexa* WT (black), *ΔflaA2* (red), and *flaA2*-complemented (blue) strains. Optical density (OD) at 420 nm was measured at 0, 24, 33, 48, 51, 72, 81, 96, and 105 h after inoculation. Measurements were terminated at 105 h (day 4), as cell sedimentation occurred after day 5. For each independent experiment, OD measurements were performed in triplicate, and the triplicate values were averaged to obtain a single data point. Data points represent the mean of three independent experiments, and error bars indicate the standard deviation. The culture medium for the *flaA2*-complemented strain contained 25 μg/mL kanamycin. (D) Immunoblotting of whole-cell lysates from WT and *ΔflaA2* mutant using anti-FcpA, anti-FlaA1, and anti-FlaA2 antisera. Whole-cell lysates were prepared in two independent experiments, and each preparation was analyzed by immunoblotting; a representative blot is shown.

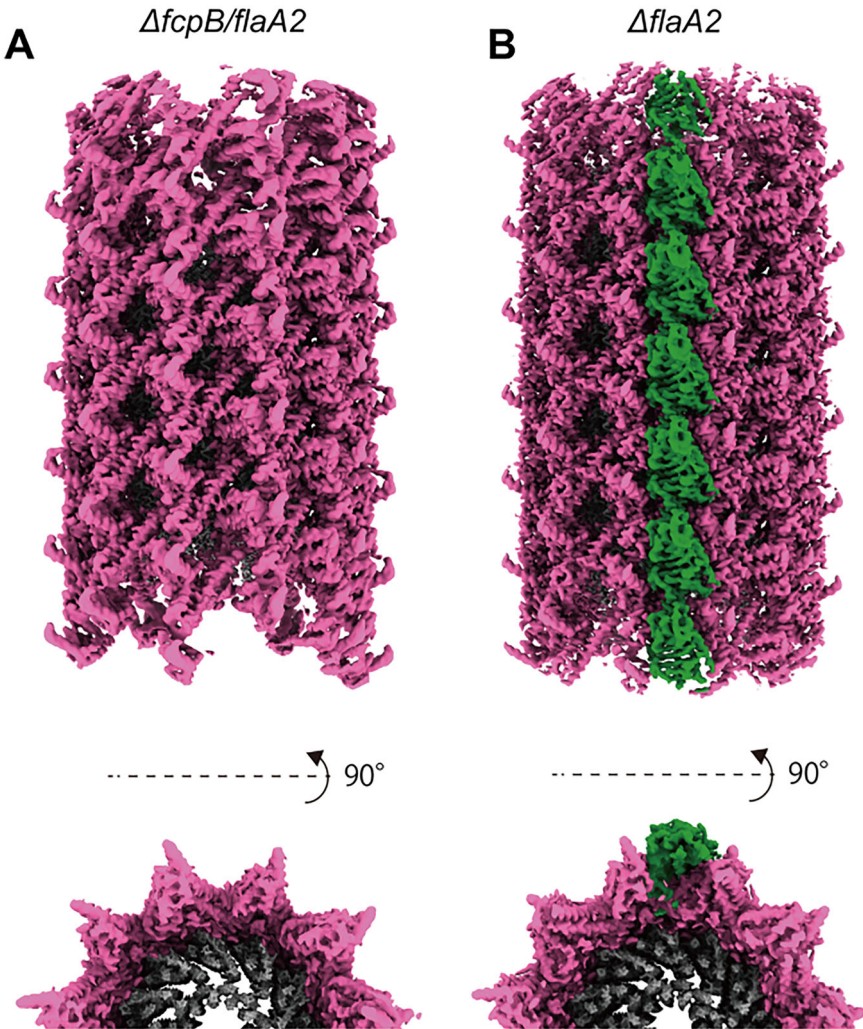

**Figure EV3.  Interaction of FcpB with FcpA.**

(A) Reconstructed image of the periplasmic flagellum (PF) from *ΔfcpB/flaA2* mutant, which is uniformly covered by FcpA alone. The FlaB core filament (gray) and FcpA (magenta) are shown. (B) Reconstructed image of the PF from *ΔflaA2* mutant, uniformly covered by both FcpA and FcpB. FlaB (gray), FcpA (magenta), and FcpB (green) are shown. See Methods for details of reconstruction reproducibility.

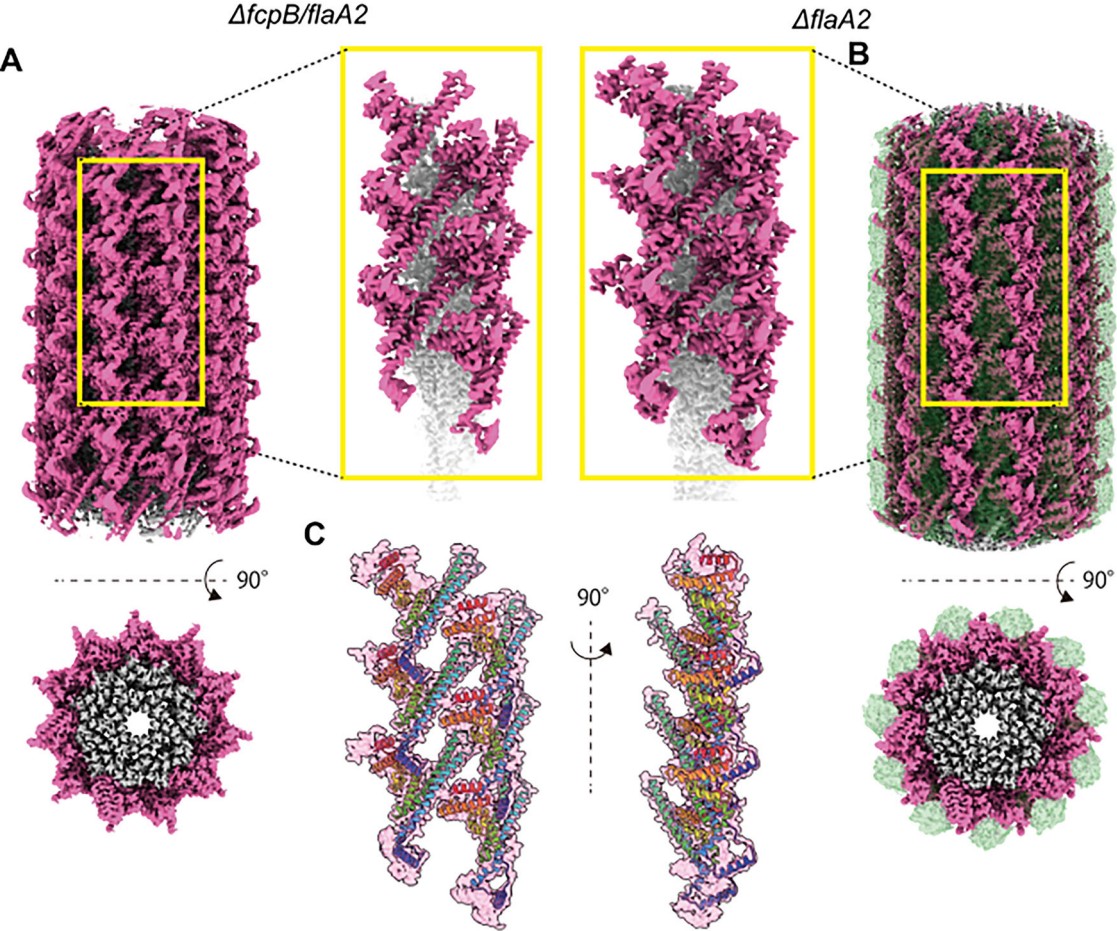

ΔfcpB/flaA2

ΔflaA2

**Figure EV4. Effect of FcpB on the structure of FcpA row.**

(**A**) Reconstructed image of the periplasmic flagellum (PF) from *ΔfcpB/flaA2* mutant, which is uniformly covered by FcpA alone. The FlaB core filament (gray) and FcpA (magenta) are shown. A magnified view of the boxed region is shown on the right. (**B**) Reconstructed image of the PF from *ΔflaA2* mutant, uniformly covered by both FcpA and FcpB. FlaB (gray), FcpA (magenta), and FcpB (green) are shown. In the magnified view (left), FcpB is omitted to highlight FcpA structure. (**C**) Structural comparison of FcpA between the two mutants. The FcpA model from *ΔfcpB/flaA2* mutant (shown as a rainbow-colored ribbon, from panel A) is superimposed onto the FcpA density map from *ΔflaA2* mutant (magenta, from panel **B**). See Methods for details of reconstruction reproducibility.

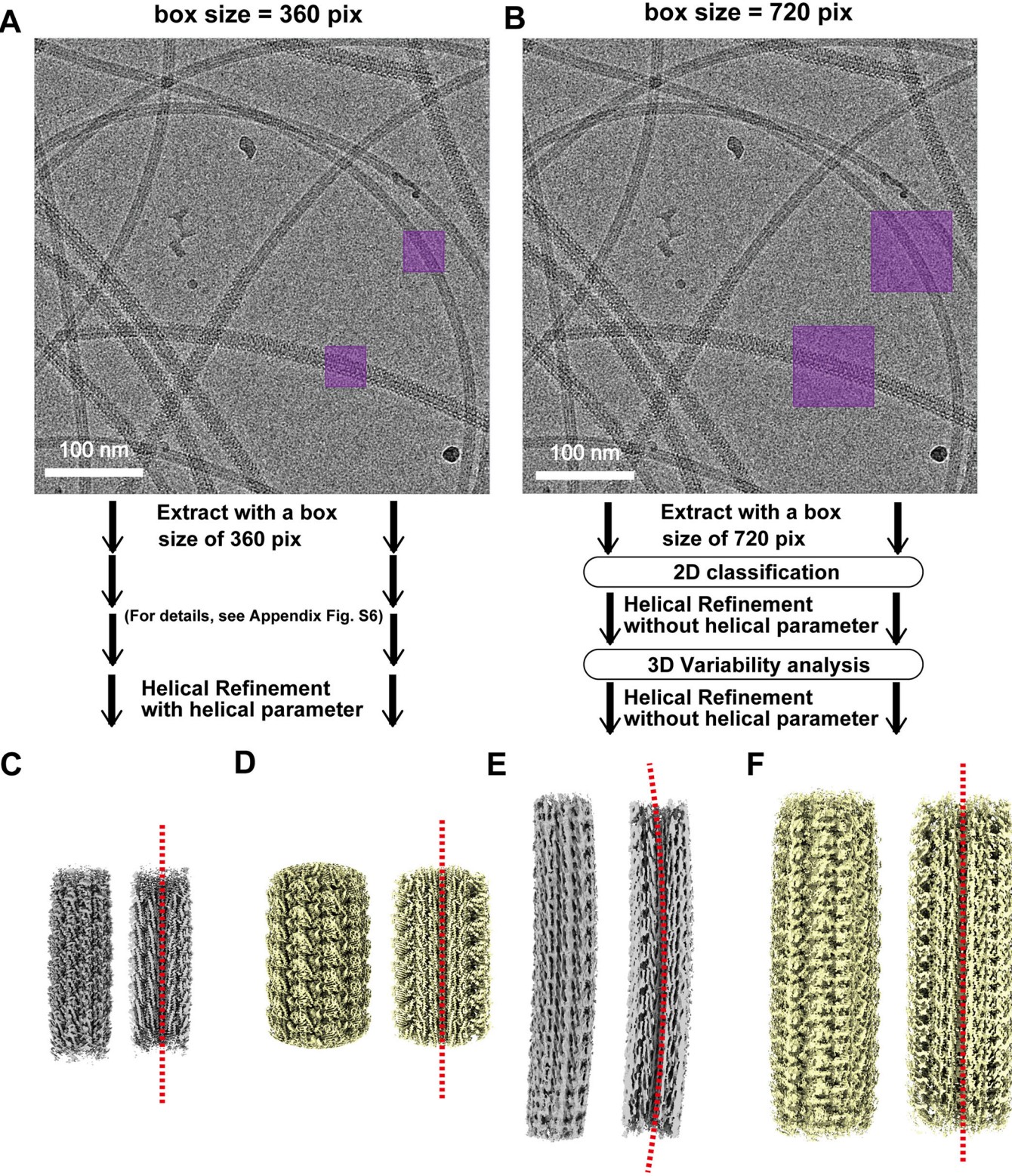

**Figure EV5. Structural comparison of Δ*flaA2* mutant PFs processed with and without applying helical symmetry.**

(A, B) Data-processing workflows for helical refinement with (A) and without (B) applying helical symmetry. The areas from which particles were extracted are indicated by purple squares. For details, see Appendix Fig. S6. (C–F) Reconstructed density maps and corresponding cross-sectional views of the core filament obtained with (C) and without (E) applying helical symmetry, and of the sheathed filament obtained with (D) and without (F) applying helical symmetry. Red dotted lines indicate the filament axis.

