## [Peer Review File · The EMBO Journal]

Asymmetric sheath coordination controls flagellar architecture and function in *Leptospira* spirochete

Akihiro Kawamoto, Toshiki Kuribayashi, Masatomo Morita, Shuichi Nakamura, and Nobuo Koizumi

Corresponding author(s): Nobuo Koizumi (koizumi.n@jihs.go.jp) , Akihiro Kawamoto (kawamoto@protein.osaka-u.ac.jp), Shuichi Nakamura (shuichi.nakamura.e8@tohoku.ac.jp)

Review Timeline:

Submission Date:	19th May 25
Editorial Decision:	7th Jul 25
Revision Received:	18th Oct 25
Editorial Decision:	11th Dec 25
Revision Received:	14th Dec 25
Accepted:	23rd Jan 26

Editor: Cornelius Schneider

Transaction Report:

Dear Dr. Koizumi,

Thank you for submitting your manuscript for consideration by the EMBO Journal. It has now been seen by three referees whose comments are shown below.

Given the referees' positive recommendations, I would like to invite you to submit a revised version of the manuscript, addressing the comments of all three reviewers. I should add that it is EMBO Journal policy to allow only a single round of revision, and acceptance of your manuscript will therefore depend on the completeness of your responses in this revised version.

Thank you for the opportunity to consider your work for publication. I look forward to your revision.

Yours sincerely,

Cornelius Schneider, PhD
Editor
The EMBO Journal
c.schneider@embojournal.org

Please remember: Digital image enhancement is acceptable practice, as long as it accurately represents the original data and conforms to community standards. If a figure has been subjected to significant electronic manipulation, this must be noted in the figure legend or in the 'Materials and Methods' section. The editors reserve the right to request original versions of figures and

the original images that were used to assemble the figure.

We realize that it is difficult to revise to a specific deadline. In the interest of protecting the conceptual advance provided by the work, we recommend a revision within 3 months (5th Oct 2025). Please discuss the revision progress ahead of this time with the editor if you require more time to complete the revisions. Use the link below to submit your revision:

Referee #1:

In this manuscript, Kawamoto and co-workers report an extensive structural and functional analysis of the periplasmic flagellum in the human pathogen *Leptospira biflexa*. They notably report the structure of its native filament, showing asymmetric binding of its two sheathed bacteria. Using an elegant combination of genetic analysis and structural studies of the flagellum filament in individual mutants, they also demonstrate that a previously-unreported protein, FlaA2, is also part of the sheath, and is required for forming a functional flagellum. Finally, they combine this data to propose a mechanism for the assembly of the periplasmic flagellum filament in spirochetes.

This is a very thorough and comprehensive study, that largely builds on the previous work from Gibson et al (eLife, 2020), but including much more advanced data. The manuscript is clear and well written, and the conclusions are carefully analysed and well supported.

While we have a few edits to suggest, to enhance the overall clarity of the manuscript and to add a few additional elements of analysis, we are supportive of its publication in EMBO.

Major changes suggested:

- Additional supplementary figure panels, showing the fit into the density of individual proteins, would be required to help convince of the author's interpretation. This is mostly critical for the flaA2-complemented strain structure, where a better view of the density attributed to FlaA2 would be required.
- Flagellar filaments are not fully helical, but rather superhelical (See Kreutzberger et al., Cell, 2022). This is very clearly the case here, judging on the micrographs. While helical reconstruction might be appropriate for structure determination with a small box size, a structure of the native filament, with no helical symmetry applied, might provide interesting insights into the curvature of the sheath proteins. Overall, comparison of the curvature of the different structures might be of interest here.
- One exception to the above is the sheath filament in the Δ flaA1/2 strain, which appears to be fully helical, at least based on the micrograph shown in Supplementary Figure 8. Is this indeed the case? If so, how does the deletion of FlaA1/2 promote helical assembly?
- Based on the structures shown in figure 3, it almost looks like FcpB adopts a position similar to the D2/D3 domains found in flagella of other bacteria (*Salmonella* for example). Some comparative analysis between these may help interpret this.
- According to supplementary Table 1, no atomic models have been built for the WT or FlaA2-complemented strains. Why is that? I would argue that these are the most important structures in this study, and would strongly encourage the authors to deposit the corresponding models and maps.

Additional minor edits:

- I recommend the authors remove the terms "Near-atomic resolution" (p3 and p11), which are not really relevant these days.
- The 'streaky' densities in Figure 2E and Figure 5C may be caused by averaging particles in different curvature, even the difference is small. Refinement from subclass of 3D classification or 3D flexible refinement may be helpful to address this.

Referee #2:

This manuscript investigates the roles of three flagellar sheath proteins - FcpB, FlaA1, and FlaA2 - in *Leptospira biflexa*. The

authors employ phenotypic assays and cryo-electron microscopy to characterize deletion mutants for each of these proteins. Key findings include the distinct roles of these sheath proteins in flagellar architecture and motility. The study shows that FlaA2, but not FlaA1, is essential for the asymmetric localization of FcpA and FcpB, which in turn is required for proper filament curvature and motility. The authors also demonstrate that FcpA induces curvature, while FcpB contributes to filament stiffness, thereby facilitating whole-cell deformation during swimming. Furthermore, they report that FlaB1, rather than FlaB2, forms the core filament, and that FlaB1 glycosylation is necessary for sheath interactions. The manuscript is clearly written, well organized, and benefits from working with clean gene deletions and high-resolution structural data. However, some conclusions are not sufficiently supported by the data presented, and key experimental details are lacking in certain sections. Addressing these points would significantly strengthen the manuscript.

Main Comments

1. Supplementary Fig. 1.: The presentation of tandem mass spectrometry data in panel B is not very informative. The authors should consider an alternative way to present this data.
2. Lines 121-123 and Fig. 1D: "Although FlaA2 was detected in purified WT PFs by immunoblotting (Sasaki et al., 2018), no corresponding density was observed in the single-particle reconstruction" - It is surprising that the authors do not detect the unidentified densities that were described in Gibson et al., 2020, especially given the improved resolution in this study.
3. Supplementary Fig. 7.: This figure is not informative without a side-by-side comparison to WT PFs. The negative staining micrographs presented elsewhere are sufficient; this experiment could be omitted or improved for clarity.
4. Lines 157-159 and Supplementary Fig. 9: The detection of FlaA1 using the Sasaki et al., 2018 method (lane 2) contradicts the claim that FlaA1 is absent. The choice to switch to a harsher purification method (1-minute vortex with glass beads) requires further justification, as it may account for the apparent absence of sheath proteins. The conclusion that "FlaA1 is not incorporated into WT PFs" is not sufficiently supported.
5. Lines 220-222: The comparison between colony growth on 0.4% agar and the soft-agar motility assay described in Wunder et al., 2018 is not directly appropriate, as colony size alone does not account for potential differences in growth rates between the $\Delta fcpB$ mutant and the WT strain. In Figure 4A, the $\Delta fcpB$ colonies appear even larger than those of the WT, which further underscores the need for quantification. To draw a meaningful conclusion, it would be important to measure colony diameters and provide growth curve comparisons between the two strains. Additionally, key methodological details such as the inoculum size and incubation times for the 0.4% agar plates are missing and should be included for clarity and reproducibility. Ideally, performing a standard soft-agar motility assay following the protocol from Wunder et al., 2018 would provide a more appropriate and interpretable comparison. As currently presented, the data do not convincingly support the conclusions drawn.
6. Fig. 4C: No information is provided on the statistical significance of the differences observed.
7. Lines 349-350: The claim that the filament core is composed solely of FlaB1 contradicts previous data (e.g., Sasaki et al., 2018, Fig. S2), which show FlaB2 as the major component in WT filaments. As your structural analysis is based on mutant strains, this discrepancy should be addressed. An RT-qPCR analysis of *flaB* isoform expression could help clarify this point.
8. All figures with quantifications: individual data points should be shown and the number of biological replicates should be stated in the figure or figure legend.

Minor Comments

1. Fig. 1: Missing scale bar in panel B.
2. Line 82: "...FlaA1 and Fla2" - missing capital "A" in FlaA2
3. Lines 142-144: "Because the two genes are arranged in an operon, the $\Delta flaA1/2$ strain lacks expression of both proteins (Supplementary Fig. 5)." - This statement, combined with the contents of Supplementary Fig. 5, is confusing. In the text, the authors claim having obtained a double knockout mutant of *flaA1* and *flaA2* genes, but in Supplementary figure 5, the immunoblot shows a strain with a single deletion of *flaA2*. The authors should revise the phrasing of this part, to make it clear that by deleting the *flaA2* gene, *flaA1* also loses its expression, therefore a *flaA2* mutant is equivalent to a double *flaA1/2* mutant. A schematic of the *flaA1-flaA2* operon with the primers used for the homologous recombination for each mutant could be useful.
4. Fig. 2A, bottom: Panels should be shown at the same scale for aesthetic consistency.
5. Line 189: The purpose of the $\Delta fcpB/flaA1/2$ strain analysis is unclear. An introductory sentence would improve readability.
6. Fig. 5B: Negatively stained purified PFs would have been more convincing than dark-field microscopy observation.
7. Lines 342-344: Redundant with lines 323-324.
8. Lines 448-455: The hypothesis of FlaA2 being associated with the protoplasmic cylinder is interesting - are there any structural cues in FlaA2 that could suggest this type of interaction?
9. Lines 550, 585: Consider writing small volumes in numerals for clarity ("2.6 μ L" instead of "Two point six microliters").
10. Line 618: "*flaA2*-complementant" - typo, should be "complemented"

Referee #3:

The research conducted by Kawamoto et al. revealed a complex flagellar assembly that influences cellular morphology, filament stiffness, and motility, utilizing the spirochete *Leptospira biflexa* as a model organism. The authors have constructed mutants in the periplasmic flagellar sheath and in genes that encode various flagellar core proteins, aiming to elucidate the assembly, rigidity, curvature, and morphology of cellular poles, as well as how these factors influence motility through protein-protein interactions and high-resolution cryo-electron microscopy (cryo-EM). Developing a systematic model was a significant

achievement, as the spirochete's internal flagella consist of multiple FlaA, FlaB, and coil proteins FcpA/B. Additionally, this study determines how the bacterial poles are bent, which is considered a significant advancement.

I have no major comments or concerns; however, I recommend that the authors consider the following suggestions to strengthen the foundation of the study and the model.

- 1) The study provides limited information about FlaA1. Can the authors offer a reasonable prediction regarding the role, location, or assembly of FlaA1? AlphaFold Multimer, utilizing FlaA1 and FlaA2, for example, may provide a predictive model.
- 2) According to Figure 7, one model suggests that FlaA2 interacts with peptidoglycan (PG). It is well established that flagellar proteins, such as MotB, bind to the cell wall through a PG-binding domain (PMID: 19820083). Is this domain conserved in FlaA2? If it is, this would strongly support the idea that FlaA2 may utilize PG as a relatively stable platform to initiate the assembly of Fcp proteins.
- 3) Filament core-sheath interactions: Figure 6 indicates that FlaB1-Thr137 is likely to interact with FliC, while FlaB1-Thr182 is associated with FcpA. Sasaki et al. demonstrated interactions between FcpA and FlaB1. The study would have been strengthened if the authors had confirmed the interactions between FcpB-FlaB1-Thr137 and FcpA-FlaB1-Thr182 using FlaB1 mutant protein derivatives.
- 4) Reword lines 18-20 to enhance clarity.
- 5) Line 34-35: Do all bacterial flagella enable motility only in liquid environments?
- 6) The labeling of the strains in Supplement Figure 5 does not match the description in lines 143-144.
- 7) Atomic models fitting the density maps: Please provide the parameters and indicate whether the fits are satisfactory.

Referee #1:

In this manuscript, Kawamoto and co-workers report an extensive structural and functional analysis of the periplasmic flagellum in the human pathogen *Leptospira biflexa*. They notably report the structure of its native filament, showing asymmetric binding of its two sheathed bacteria. Using an elegant combination of genetic analysis and structural studies of the flagellum filament in individual mutants, they also demonstrate that a previously-unreported protein, FlaA2, is also part of the sheath, and is required for forming a functional flagellum. Finally, they combine this data to propose a mechanism for the assembly of the periplasmic flagellum filament in spirochetes.

This is a very thorough and comprehensive study, that largely builds on the previous work from Gibson et al (eLife, 2020), but including much more advanced data. The manuscript is clear and well written, and the conclusions are carefully analysed and well supported.

While we have a few edits to suggest, to enhance the overall clarity of the manuscript and to add a few additional elements of analysis, we are supportive of its publication in EMBO.

We sincerely thank the referee for the time and effort devoted to reviewing our manuscript, and for the positive and supportive feedback. We have carefully addressed all of the referee's specific comments, and we believe that the revisions have improved the clarity and quality of the manuscript, fully addressing the concerns raised.

Comment 1: Additional supplementary figure panels, showing the fit into the density of individual proteins, would be required to help convince of the author's interpretation. This is mostly critical for the *flaA2*-complemented strain structure, where a better view of the density attributed to FlaA2 would be required.

Reply: Thank you for this helpful comment. In response, we have prepared additional figures showing the fitted models of individual proteins (FlaB1, FlaB2, FcpA, and FcpB) and their corresponding density maps (Fig. EV1 and Appendix Fig. S6). These figures illustrate how each atomic model fits into the reconstructed densities for the *ΔflaA2* strain, in which even side chains are well resolved, as well as for the WT. As shown below, unidentified densities form continuous rows located at the edge of the FcpA/FcpB sheath. Unfortunately, even after reanalysis, we were unable to fit FlaA2

into these additional densities observed only in the *flaA2*-complemented strain, as they were not sufficiently resolved for reliable model building. As described in the text, FlaA2 in the WT may be lost during purification, and its binding to the core filament may be weak or transient in nature. Therefore, while the density attributable to FlaA2 cannot be conclusively modeled, its unique appearance in the complemented strain is most consistent with being derived from FlaA2.

Comment 2: Flagellar filaments are not fully helical, but rather superhelical (See Kreutzberger et al., Cell, 2022). This is very clearly the case here, judging on the micrographs. While helical reconstruction might be appropriate for structure determination with a small box size, a structure of the native filament, with no helical symmetry applied, might provide interesting insights into the curvature of the sheath proteins. Overall, comparison of the curvature of the different structures might be of interest here.

Reply: Thank you for the referee's insightful and important comment. We agree that native flagellar filaments are not perfectly helical but exhibit superhelical curvature, as also observed in our micrographs. To address this point, we reconstructed the $\Delta flaA1/flaA2$ (= $\Delta flaA2$ in the revised manuscript) and $\Delta fcpB$ ($\Delta fcpB_{CL15}$) filaments both with and without helical refinement, using two different box sizes (360 and 720 pixels), as shown below.

While the smaller box size (360 pixels) captured only local helical symmetry, analysis

with the larger box size (720 pixels) revealed that the filaments exhibit superhelical curvature. In both mutants, the reconstructions without helical refinement for thin filaments (= core filaments) preserved the overall curvature, supporting the notion that the core filaments are superhelical rather than perfectly helical. In contrast, the curvature of fully sheathed (thick) filaments was smaller than that of the core filaments, indicating that the sheath contributes to the rigidity of the periplasmic flagella. Conversely, the asymmetric localization of the sheath observed in the wild type appears to maintain the superhelical shape of the core filaments, which is essential for the generation of functional periplasmic flagella.

We have included Fig. EV5 and Appendix Fig. S14, prepared from these analyses, described the analysis procedures in the Methods and Protocols section (P16L570), and added the following sentences in the Discussion section:

P10L372. Notably, flagellar filaments form superhelical structures not only in bacteria but also in archaea (Kreutzberger *et al.*, 2022). In spirochetal *Leptospira*, we observed a superhelical configuration in unsheathed core filaments (Fig. EV5; Appendix Fig. S14). In contrast, the uniformly sheathed PFs from both *ΔflaA2* and *ΔfcpB* mutants exhibited markedly smaller curvature than the core filaments (Fig. EV5; Appendix Fig. S14). Because FcpA contributes to coiling formation while FcpB provides rigidity, the uniform sheath likely suppressed the intrinsic superhelical curvature of the core

filaments. These findings suggest that the asymmetric localization of the sheath in the wild type maintains the superhelical shape of the core filaments, which is essential for the generation of functional PFs.

Comment 3: One exception to the above is the sheath filament in the Δ flaA1/2 strain, which appears to be fully helical, at least based on the micrograph shown in Supplementary Figure 8. Is this indeed the case? If so, how does the deletion of FlaA1/2 promote helical assembly?

Reply: Thank you for the referee's insightful comment. As described above, the curvature of fully sheathed filaments in the FlaA1/FaA2-deficient mutant was smaller than that of the core filaments. Deletion of *flaA2* caused loss of the asymmetric localization of the sheath proteins FcpA and FcpB (Fig. 2), resulting in uniform coverage of the core by these proteins. This symmetric sheath arrangement likely restricts the intrinsic superhelical curvature of the core filaments, which makes the filaments appear nearly fully helical in the micrographs.

Comment 4: Based on the structures shown in figure 3, it almost looks like FcpB adopts a position similar to the D2/D3 domains found in flagella of other bacteria (Salmonella for example). Some comparative analysis between these may help interpret this.

Reply: Thank you for the referee's comment. In response, we compared the structures of *Salmonella* FliC (light blue) with the FlaB1 (gray)/FcpA (magenta)/FcpB (green) complex, as shown below. The alignment was performed based on the D0/D1 domains of FliC and FlaB1, which overlapped well. However, neither FcpB nor FcpA aligned with the D2/D3 domains of FliC, and their orientations were clearly different, indicating that the functions of FcpB (and FcpA) are unrelated to those of the D2/D3 domains of flagellin.

Comment 5: According to supplementary Table 1, no atomic models have been built for the WT or FlaA2-complemented strains. Why is that? I would argue that these are the most important structures in this study, and would strongly encourage the authors to deposit the corresponding models and maps.

Reply: Thank you for this important comment. We apologize for the confusion caused by the incomplete supplementary table submitted with the initial version of the manuscript. At the time of submission, the atomic models for both the WT and *flaA2*-complemented strains had already been built; however, deposition to the public databases had not yet been completed, and as a result, the supplementary table was submitted in an incomplete form. Our intention had been to update the table prior to peer review, but this was unfortunately not achieved in time.

In the revised manuscript, we have now included the completed supplementary table (Appendix Table 3) and confirm that the corresponding cryo-EM maps and atomic models for both WT and *flaA2*-complemented strains have been deposited.

Comment 6: I recommend the authors remove the terms "Near-atomic resolution" (p3 and p11), which are not really relevant these days.

Reply: Thank you for the referee's comment. In response, we have removed the term 'near-atomic resolution' from the revised manuscript as follows:

(Original) Here, we generated sheath protein knockout mutants and used cryo-electron microscopy at near-atomic resolution to elucidate the mechanisms underlying PF assembly, curvature, and rigidity in *Leptospira biflexa*.

(Revised: P1L21) Here, we generated sheath protein knockout mutants and used high-resolution cryo-electron microscopy at near-atomic resolution to elucidate the mechanisms underlying PF assembly, curvature, and rigidity in *Leptospira biflexa*.

(Original) By integrating these phenotypic analyses with a near-atomic resolution structure obtained by cryo-EM, we propose distinct and cooperative roles for each sheath protein in shaping the unique architecture of the *Leptospira* flagellum.

(Revised: P3L113) By integrating these phenotypic analyses with structural insights from cryo-EM, we propose distinct and cooperative roles for each sheath protein in shaping the unique architecture of the *Leptospira* flagellum.

(Original) In contrast, the PFs purified from the $\Delta fcpB$ _KO15 strain (sheathed uniformly with FcpA) and the $\Delta flaA1/2$ strain (sheathed uniformly with both FcpA and FcpB) are straight, enabling the acquisition of near-atomic resolution structures (Supplementary Figs. 8 and 15).

(Revised: P6L226) In contrast, the PFs purified from the $\Delta fcpB$ _KO15 strain (sheathed uniformly with FcpA) and the $\Delta flaA1/2$ strain (sheathed uniformly with both FcpA and FcpB) are straight, enabling the acquisition of high-resolution structures (Appendix Figs. S6 and S12).

Comment 7: The 'streaky' densities in Figure 2E and Figure 5C may be caused by averaging particles in different curvature, even the difference is small. Refinement from subclass of 3D classification or 3D flexible refinement may be helpful to address this.

Reply: Thank you for the referee's insightful comment. We acknowledge that the streaky densities observed in Figures 2E and 5C may, at least in part, arise from heterogeneity in filament curvature. In response, we attempted 3D classification of particles based on curvature. However, with the current dataset, subclassification did not yield well-separated particle groups and did not improve the reconstructions. Flexible refinement approaches were also considered, but the limited number of particles precluded their effective application.

We have therefore retained the present reconstructions, while noting in the revised manuscript that curvature heterogeneity may contribute to the observed densities. We

recognize this as an important methodological challenge and will continue to explore improved strategies to address this limitation in future studies.

In the results section, we have added the following sentence:

P5L158. Although these structures could not be conclusively identified due to limited resolution, they differed from FcpA and FcpB and are therefore likely to represent FlaA2. The streaky densities observed in these reconstructions may, at least in part, reflect curvature heterogeneity of the filaments.

Referee #2:

This manuscript investigates the roles of three flagellar sheath proteins - FcpB, FlaA1, and FlaA2 - in *Leptospira biflexa*. The authors employ phenotypic assays and cryo-electron microscopy to characterize deletion mutants for each of these proteins. Key findings include the distinct roles of these sheath proteins in flagellar architecture and motility. The study shows that FlaA2, but not FlaA1, is essential for the asymmetric localization of FcpA and FcpB, which in turn is required for proper filament curvature and motility. The authors also demonstrate that FcpA induces curvature, while FcpB contributes to filament stiffness, thereby facilitating whole-cell deformation during swimming. Furthermore, they report that FlaB1, rather than FlaB2, forms the core filament, and that FlaB1 glycosylation is necessary for sheath interactions. The manuscript is clearly written, well organized, and benefits from working with clean gene deletions and high-resolution structural data. However, some conclusions are not sufficiently supported by the data presented, and key experimental details are lacking in certain sections. Addressing these points would significantly strengthen the manuscript.

We sincerely thank the referee for the time and effort devoted to reviewing our manuscript, as well as for the constructive and insightful feedback. We have carefully considered all of the referee's comments and addressed each point in detail. In revising the manuscript, we have clarified key experimental details, modified statements where the original conclusions were not fully supported by the data, and added additional explanation where needed. We believe that these revisions have improved both the clarity and the overall quality of the manuscript, and that the concerns raised have been fully addressed.

Comment 1: Supplementary Fig. 1.: The presentation of tandem mass spectrometry data in panel B is not very informative. The authors should consider an alternative way to present this data.

Reply: Thank you for the referee's comment. In response to the concern that the original panel B was not sufficiently informative, we have revised the figure and its accompanying data presentation. The updated Supplementary Figure 1B (Appendix Fig. S1) now shows a schematic representation of the FcpB protein with the precise positions of the peptides identified by MALDI-TOF-MS/MS. In addition, we have added Appendix Table S1, which provides detailed information for each peptide, including its position within the protein, mass accuracy (ppm), Mascot score, and whether it uniquely matches the target protein. We believe these revisions substantially improve the clarity and interpretability of the MS data.

Comment 2: Lines 121-123 and Fig. 1D: "Although FlaA2 was detected in purified WT PFs by immunoblotting (Sasaki et al., 2018), no corresponding density was observed in the single-particle reconstruction" - It is surprising that the authors do not detect the unidentified densities that were described in Gibson et al., 2020, especially given the improved resolution in this study.

Reply: Thank you for the referee's comment. As stated in the original manuscript, we were not able to observe density corresponding to FlaA2 in WT PFs, which may have dissociated during purification. For immunoblotting, PFs were purified without sucrose density gradient ultracentrifugation, whereas cryo-EM samples were prepared using sucrose density gradient ultracentrifugation to obtain clearer reconstructions. This difference in sample preparation may partly account for the absence of FlaA2 density in our cryo-EM data.

In addition, as noted in the Discussion section of the original manuscript, a previous study reported that FlaA2-associated core filament structures were observed in fewer than 5% of the analyzed particles, and in 60–80% of cases no additional density was observed on the core filament. This very low frequency suggests that FlaA2 binding to PFs may be inherently weak or transient.

To clarify the difference in sample preparation between immunoblotting and cryo-EM analysis, we have revised the descriptions in the Materials and Methods section as follows:

P14L492. For cryo-EM analysis, PFs were further purified using a 20–50%(w/w) continuous sucrose density gradient in buffer C (10 mM Tris-HCl pH 8.0, 5 mM EDTA-NaOH, 1% Triton X-100), centrifuged in a swing rotor at $49,100 \times g$ for 18 h at 4 °C.

P14L500. Protein samples (1.5 µg of purified PFs without sucrose density gradient ultracentrifugation or lysates from 1.5×10^8 leptospiral cells) were prepared in SDS-PAGE sample buffer and separated on 10% SDS-PAGE gels, followed by Western blotting.

As a future direction, targeted cryo-EM analyses and complementary labeling approaches will be required to directly visualize the stability and spatial distribution of FlaA2 along the filament, as described in the Discussion section of the revised manuscript (please see our reply to Comment 4).

Comment 3: Supplementary Fig. 7.: This figure is not informative without a side-by-side comparison to WT PFs. The negative staining micrographs presented elsewhere are sufficient; this experiment could be omitted or improved for clarity.

Reply: We agree with the referee that, in this context, Supplementary Fig. 7 is not sufficiently informative without side-by-side images of WT PFs. In the revised version, we have therefore added dark-field images of WT PFs to Supplementary Fig. 7 (Appendix Fig. S5).

Regarding the referee's point that the negative-staining micrographs are sufficient (also related to Comment 14), we would like to clarify our rationale. While differences in PF curvature can indeed be observed in the negatively stained electron micrographs (Fig. 2E), negative staining often introduces artifacts due to drying and surface tension, and thus does not necessarily preserve the native morphology of biological samples.

In this study, we aimed to provide quantitative analysis of PF morphology (Fig. 5), and for that reason we employed dark-field microscopy to observe isolated PFs in aqueous solution. Although dark-field imaging is widely applied for external flagella in *E. coli*, *Salmonella*, and *Vibrio*, it has rarely been reported for periplasmic flagella of spirochetes. To our knowledge, a similar approach was only used by Li et al. (J. Bacteriol, 2008) to measure PF stiffness in *Brachyspira*.

For these reasons, we believe it is critical to present PF curvature data obtained by dark-field microscopy, and we hope the referee will recognize the added value of these

results.

Comment 4: Lines 157-159 and Supplementary Fig. 9: The detection of FlaA1 using the Sasaki et al., 2018 method (lane 2) contradicts the claim that FlaA1 is absent. The choice to switch to a harsher purification method (1-minute vortex with glass beads) requires further justification, as it may account for the apparent absence of sheath proteins. The conclusion that "FlaA1 is not incorporated into WT PFs" is not sufficiently supported.

Reply: Thank you for the referee's comment. We agree that we cannot definitively assert that FlaA1 is not incorporated into WT PFs. In this study, we employed a different extraction method to obtain clearer cryo-EM reconstructions. As noted above, FlaA2 was not detected in WT PFs extracted using this method, suggesting that FlaA1 may also have been dissociated during purification. We have therefore revised the relevant sentence to indicate the possibility of dissociation during purification, as follows:

(Previous) Although FlaA1 was previously detected in PFs using a different extraction method (Sasaki et al., 2018), repeating that method failed to detect FlaA1, suggesting that FlaA1 is not incorporated into WT PFs (Supplementary Fig. 9).

(Revised: P4L148) Although FlaA1 was previously detected in PFs using a different extraction method (Sasaki et al., 2018), repeating that method failed to detect FlaA1, suggesting that FlaA1 is not stably associated with the core filament and/or other sheath proteins, and may have dissociated during purification (Appendix Fig. S7).

As noted for FlaA2, similar targeted cryo-EM and complementary labeling strategies will also be required to determine the stability and spatial distribution of FlaA1 along the PF.

We have also added the following sentences to the Discussion section to clarify the role of FlaA1 (please see also our reply to Comment 1 from referee #3):

P9L329. This study also demonstrated that complementation with *flaA2* alone restored the phenotypic and functional defects in the FlaA1/FlaA2-deficient mutant, whereas the FlaA1-deficient mutant exhibited colony sizes comparable to the WT strain (Fig. 2 and

Appendix Fig. S4). FlaA1 is predicted to contain a signal peptide, similar to FlaA2, suggesting secretion via the Sec/SPI pathway. Both proteins possess the FlaA domain, and their predicted tertiary structures are highly similar, as confirmed by pairwise structural alignment (TM scores: 0.52 overall; 0.58 for the FlaA domain). However, multimer predictions using AlphaFold 3 and AlphaFold2-Multimer yielded ipTM scores of 0.37 and 0.57, respectively, which do not support a stable interaction. Thus, while structurally similar, FlaA1 and FlaA2 are unlikely to form a heteromeric complex. As with FlaA2, FlaA1 may also dissociate from PFs during purification. Future work will require targeted cryo-EM analyses and complementary labeling approaches to directly visualize the stability and spatial distribution of both FlaA2 and FlaA1 along the PF.

Comment 5: Lines 220-222: The comparison between colony growth on 0.4% agar and the soft-agar motility assay described in Wunder et al., 2018 is not directly appropriate, as colony size alone does not account for potential differences in growth rates between the $\Delta fcpB$ mutant and the WT strain. In Figure 4A, the $\Delta fcpB$ colonies appear even larger than those of the WT, which further underscores the need for quantification. To draw a meaningful conclusion, it would be important to measure colony diameters and provide growth curve comparisons between the two strains. Additionally, key methodological details such as the inoculum size and incubation times for the 0.4% agar plates are missing and should be included for clarity and reproducibility. Ideally, performing a standard soft-agar motility assay following the protocol from Wunder et al., 2018 would provide a more appropriate and interpretable comparison. As currently presented, the data do not convincingly support the conclusions drawn.

Reply: Thank you for the referee's comment. We agree that photographs without quantification were not sufficient for comparison, and we apologize for the lack of methodological details. In response, we have performed additional experiments to address these concerns.

First, we compared the growth of all strains under the experimental conditions described in P12L439. The $\Delta fcpB$ mutants ($\Delta fcpB_{CL13}$ and $\Delta fcpB_{CL15}$) grew slightly more slowly than the WT strain, whereas the $\Delta flaA2$ and $\Delta flaA1$ mutants exhibited growth comparable to WT. Growth curves for these strains are now provided in the revised manuscript (Figure EV2 and Appendix Figs. S2 and S4).

Second, we repeated the colony formation assay described in the original manuscript, this time measuring colony diameters on day 7 after inoculation. Colony sizes were quantified at pixel level using ImageJ (described in P13L454). As a results, the

*ΔfcpB*_CL13 strain formed colonies that were statistically smaller than those of WT but larger than those of the *ΔfcpB*_CL15 strain (Fig. 4 and Appendix Fig. S2).

Finally, as suggested by the referee, we tested the soft-agar motility assay following Wunder et al. (2018). However, we found that variations in the inoculation spot substantially affected the results. Therefore, we adopted an alternative motility assay in which wells were created in EMJH soft agar plates, and 2 μL of bacterial suspension were added into each well (described in P12L449). The spread of cells in the agar was then quantified at pixel level using ImageJ as mentioned above. This assay yielded results consistent with the colony size comparisons (Appendix Fig. S10).

In light of these additional experimental results, we have revised Fig. 4 and the accompanying text as follows:

P5L184. FcpB is required for efficient motility under low-viscosity conditions. The *ΔfcpB*_CL13 strain, constructed via allelic exchange, formed significantly smaller colonies than the WT strain on 0.4% agar plates ($P = 2.73 \times 10^{-77}$; Fig. 4A,B). This phenotype, consistent with the reduced colony spreading reported by Wunder et al. (2018), is attributable to the loss of FcpB, as confirmed by immunoblotting of purified PFs from the mutant (Fig. 4C). In viscous medium, Wunder et al. observed only a slight motility defect upon *fcpB* deletion; in agreement, our *ΔfcpB*_CL13 strain swam somewhat more slowly than WT in 0.5% methylcellulose (MC), swimming at ~70% of the WT speed ($P = 0.18$; Fig. 4D). The soft-agar motility assay likewise revealed reduced motility under high-viscosity conditions (Appendix Fig. S10). Growth analysis further showed that the *ΔfcpB*_CL13, as well as another *ΔfcpB* mutant (*ΔfcpB*_CL15), grew modestly more slowly than the WT strain (Appendix Fig. S2). Together with the reduced swimming speed, this slower growth may have contributed to the reduced colony size and spreading of the *ΔfcpB* mutant on 0.4% agar. Both WT and *ΔfcpB*_CL13 strains exhibited higher swimming speeds in 0.5% MC compared with 0% MC (Fig. 4D). In contrast, *ΔfcpB*_CL13 displayed markedly reduced motility in motility buffer without MC (0% MC), swimming at ~15% of the WT speed ($P = 3.62 \times 10^{-13}$; Fig. 4D). Thus, while *ΔfcpB*_CL13 retained its ability to respond to increased viscosity, its motility was severely impaired under low-viscosity conditions, highlighting that FcpB is critical for efficient motility under low-viscosity conditions.

In addition, we also quantified the colony sizes of the *ΔflaA2* mutant, its complemented strain, and the *ΔflaA1* mutant, as described above. The colony size of the *ΔflaA2*-

complemented strain was statistically smaller than that of WT but larger than that of *AflaA2* (Figure EV2B). The complemented strain grew more slowly than WT in the presence of kanamycin (Figure EV2C), which likely accounts for its smaller colony size, as its swimming speed was comparable to that of WT. By contrast, both the growth and colony size of *AflaA1* were comparable to those of WT (Appendix Fig. S4).

We have added the following sentence in P5L154:

“The colony size of the complemented strain was smaller than that of the WT, likely due to slower growth in the presence of kanamycin, as its swimming speed was comparable to that of the WT (Fig. 2B; Fig. EV2).”

Comment 6: Fig. 4C: No information is provided on the statistical significance of the differences observed.

Reply: Thank you for pointing this out. We apologize for not including statistical analyses for Fig. 4C in the original submission. In the revised manuscript, we have now added the results of statistical analyses not only for Fig. 4C (corresponding to Fig. 4D in the revised version) but also for Fig. 2B. Statistical significance was assessed using Student's *t*-test, and the results are now clearly indicated in the respective figure legends.

Comment 7: Lines 349-350: The claim that the filament core is composed solely of FlaB1 contradicts previous data (e.g., Sasaki et al., 2018, Fig. S2), which show FlaB2 as the major component in WT filaments. As your structural analysis is based on mutant strains, this discrepancy should be addressed. An RT-qPCR analysis of flaB isoform expression could help clarify this point.

Reply: Thank you for the referee's comment. We would like to clarify that we do not claim that the core filament is composed solely of FlaB1. In the present study, we found that unsheathed core filaments in mutant strains with straight PFs were composed exclusively of FlaB2, whereas sheathed filaments consisted exclusively of FlaB1 (Appendix Figs. 8 and 12). In our previous study (Sasaki et al., 2018), we reported that the WT strain expressed both FlaB1 and FlaB2, both of which were incorporated into PFs. The purified PF fraction from WT contains a mixture of sheathed and unsheathed PFs, and thus the previous findings are not contradictory to our current results. In that study, the amount of FlaB2 in PFs was greater than that of FlaB1, whereas whole-cell

immunoblotting showed comparable levels of FlaB1 and FlaB2. Notably, in the sheath-protein-deficient strain (*ΔfcpA*), only FlaB2 was detected in PFs, suggesting that FcpA stabilizes FlaB1-based filaments. This instability of unsheathed FlaB1 cores may account for the reduced incorporation of FlaB1 into PFs compared with FlaB2.

To further address the referee's concern, we performed RT-qPCR to compare the expression levels of *flaB1* and *flaB2* in WT, *ΔfcpB*_CL15, and *ΔflaA2* strains. Specific primers for *flaB1* (forward: 5'-CCAAGTCGAGGTTTCACAGC-3'; reverse: 5'-TAGCAGTTGGGTTGAGTCGA-3') and *flaB2* (forward: 5'-ATCGGAACTTTGGATGCTGC-3'; reverse: 5'-AGCGTTCATGAGCCCTTTTG-3') showed amplification efficiencies of 93.6% and 94.0%, respectively. Total RNA was extracted from cells at approximately OD₄₂₀ = 0.2 using the RNeasy Plus Mini Kit (Qiagen), followed by cDNA synthesis with the ReverTra Ace qPCR RT Master Mix with gDNA Remover (TOYOBO). qPCR was carried out using Thunderbird SYBR qPCR Mix (TOYOBO) under the following conditions: 95 °C for 60 s, followed by 40 cycles of 95 °C for 10 s and 60 °C for 60 s, and melting curve analysis. RNA extraction/cDNA synthesis and qPCR were performed in duplicate. $\Delta\Delta\text{Ct}$ analysis, using *flaB1* as the reference gene, was used to assess the relative *flaB2* expression in each mutant compared to WT, with the following results:

Experiment	Strain	Cq flaB1	Cq flaB2	ΔCt^*	$\Delta\Delta\text{Ct}^\#$	Fold induction
1st	WT	20.28	19.57	-0.71	0.00	1.00
	ΔfcpB _KO15	19.56	19.17	-0.39	0.32	0.80
	ΔflaA2	20.16	19.66	-0.50	0.21	0.86
2nd	WT	19.81	19.2	-0.61	0.00	1.00
	ΔfcpB _KO15	20.1	19.83	-0.32	0.29	0.82
	ΔflaA2	20.8	20.68	-0.12	0.49	0.71

* $\Delta\text{Ct} = \text{Cq}_{flaB2} - \text{Cq}_{flaB1}$

$^\# \Delta\Delta\text{Ct} = \Delta\text{Ct}_{\text{sample}} - \Delta\text{Ct}_{\text{WT}}$

These results indicate that the relative expression levels of *flaB2* and *flaB1* are similar across the tested strains, supporting that the observed filament composition differences are not due to differential transcription of the *flaB* isoforms.

Comment 8: All figures with quantifications: individual data points should be shown and the number of biological replicates should be stated in the figure or figure legend.

Reply: Thank you for the referee's helpful comment. In response, we have revised the figures to display all individual data points (Figs. 2B, 4B, 4D, and 5D, as well as Fig. EV2B and Appendix Figs. S4C and S10B). We have also specified the number of biological replicates in the corresponding figure legends, as well as in other relevant legends, to ensure clarity and reproducibility.

Comment 9: Fig. 1: Missing scale bar in panel B.

Reply: Thank you for pointing this out. We apologize for the oversight and have now added a scale bar to panel B in Fig. 1 of the revised manuscript.

Comment 10: Line 82: "...FlaA1 and Fla2" - missing capital "A" in FlaA2

Reply: Thank you for pointing out the typo. We have corrected "Fla2" to "FlaA2" in the revised manuscript.

Comment 11: Lines 142-144: "Because the two genes are arranged in an operon, the Δ flaA1/2 strain lacks expression of both proteins (Supplementary Fig. 5)." - This statement, combined with the contents of Supplementary Fig. 5, is confusing. In the text, the authors claim having obtained a double knockout mutant of *flaA1* and *flaA2* genes, but in Supplementary figure 5, the immunoblot shows a strain with a single deletion of *flaA2*. The authors should revise the phrasing of this part, to make it clear that by deleting the *flaA2* gene, *flaA1* also loses its expression, therefore a *flaA2* mutant is equivalent to a double *flaA1/2* mutant. A schematic of the *flaA1*-*flaA2* operon with the primers used for the homologous recombination for each mutant could be useful.

Reply: Thank you for the referee's helpful comment. First, we apologize for the confusion regarding the knockout mutant. In fact, we generated a Δ *flaA2* mutant by homologous recombination, not a double knockout of *flaA1* and *flaA2*. Since these two genes are arranged in an operon with *flaA2* upstream of *flaA1*, deletion of *flaA2* resulted in the loss of FlaA1 expression, as shown in Supplementary Fig. 5 of the original manuscript (corresponding to Figure EV2 in the revised manuscript). To clarify this, we have revised the following sentences in the manuscript and now consistently refer to the mutant as "*ΔflaA2*" instead of "*ΔflaA1/2*":

(Original) To investigate the roles of FlaA proteins in the assembly of coiled *Leptospira*

PF, we generated a *ΔflaA1* single knockout and a *ΔflaA1/2* double knockout strain by homologous recombination. Because the two genes are arranged in an operon, the *ΔflaA1/2* strain lacks expression of both proteins (Supplementary Fig. 5).

(Revised: P4L132) To investigate the roles of FlaA proteins in the assembly of coiled *Leptospira* PF, we generated *ΔflaA1* and *ΔflaA2* knockout strains by homologous recombination. Because the two genes are arranged in an operon, the *ΔflaA2* strain lacks expression of both proteins (Fig. EV2).

Additionally, to clarify the genetic organization and experimental design, we have included a schematic diagram of the *flaA2–flaA1* operon (genomic positions 2407400–2410200 of *L. biflexa* serovar Patoc strain Patoc I (Paris); GenBank accession no. CP000786.1), showing the positions of the primers (P1–P4) used for the construction of the *ΔflaA2* mutant in Figure EV2. We have also added a schematic diagram showing the primer positions used for generating the *ΔflaA1* mutant to Appendix Fig. S4.

Comment 12: Fig. 2A, bottom: Panels should be shown at the same scale for aesthetic consistency.

Reply: Thank you for the referee's comment. We apologize for the oversight in the initial submission. Upon re-examining the original data, we realized that the EM images had been recorded at different magnifications (2K for WT and *ΔflaA2*, and 1K for the complemented strain). In assembling the original figure, we mistakenly used the corresponding scale bars directly, resulting in an inconsistency. In the revised manuscript, we have corrected this error by applying the appropriate scale bars and adjusting all panels to the same scale, as suggested.

Comment 13: Line 189: The purpose of the *ΔfcpB/flaA1/2* strain analysis is unclear. An introductory sentence would improve readability.

Reply: Thank you for the referee's comment. In response, we have added an introductory sentence to improve readability, as shown below. We also apologize for omitting the strain name (*ΔflaA2*) in the original manuscript. Although its PF structure was analyzed and shown in Fig. 2D, this was not explicitly stated in the text.

(Previous) **FcpB stabilizes FcpA structure.** The 2.37 Å resolution structure of PFs

from the *ΔfcpB/flaA1/2* strain showed that FcpA and FcpB formed separate rows, each composed exclusively of FcpA or FcpB, respectively (Figs. 2D, 3A, and 3B).

(Revised: P5L166) **FcpB stabilizes FcpA structure.** Due to the inherent curvature of WT PFs, obtaining high-resolution cryo-EM reconstructions is challenging. To enable high-resolution structural analysis, we therefore examined straight PFs from the *ΔflaA2* and *ΔfcpB/flaA2* strains. The 2.37 Å resolution structure of PFs from the *ΔfcpB/flaA2* strain, along with the PF structure from the *ΔflaA2* strain described above, showed that FcpA and FcpB formed separate rows, each composed exclusively of FcpA or FcpB, respectively (Figs. 2D, 3A, and 3B).

Comment 14: Fig. 5B: Negatively stained purified PFs would have been more convincing than dark-field microscopy observation.

Reply: As we noted in our response to Comment 3, to conduct the quantitative analyses shown in Figures 5D and 5E, it was critical to observe PFs in their native conformation in aqueous solution, which can be achieved with dark-field microscopy but not with negatively stained samples. Negative staining often introduces artifacts due to drying and surface tension, which can distort PF morphology and prevent reliable curvature measurements. For this reason, we believe it is essential to include the dark-field microscopy data, and we kindly ask the referee to take this into consideration when evaluating our approach.

Comment 15: Lines 342-344: Redundant with lines 323-324.

Reply: Thank you for the referee's comment. We have deleted the sentence describing the number and essentiality of *flaB* genes in *B. hyodysenteriae* from the revised manuscript to avoid redundancy with lines 250–251 (corresponding to the previous lines 323–324).

Comment 16: Lines 448-455: The hypothesis of FlaA2 being associated with the protoplasmic cylinder is interesting - are there any structural cues in FlaA2 that could suggest this type of interaction?

Reply: Thank you for the referee's comment. To evaluate whether FlaA2 contains a conserved peptidoglycan-binding domain, we performed pairwise structural alignment

between FlaA2 and the peptidoglycan-binding domain of MotB from *E. coli* (UniProt ID: P0AF06) using the RCSB Protein Data Bank alignment tool with AlphaFold-predicted structures. The resulting pTM score was 0.36, indicating no significant structural similarity. Furthermore, the known peptidoglycan-association motif (NX₂LSX₂RAX₂VX₃L) was not identified in the FlaA2 sequence. These results suggest that FlaA2 does not possess a canonical peptidoglycan-binding domain, and its mechanism for associating with peptidoglycan remains to be elucidated.

We have added the following sentences to the Discussion section:

P11L402. The third (bottom) model proposes that FlaA2 first associates with the PC and forms “rails” that guide subsequent FlaB1 polymerization along a specific trajectory. This could provide spatial cues for curvature of the growing filament. However, FlaA2 lacks both a conserved peptidoglycan-binding domain—present in OmpA-related outer membrane proteins and MotB—and the known peptidoglycan-association motif (NX₂LSX₂RAX₂VX₃L) (DeMot and Vanderleyden, 1994; Koebnik, 1995). The mechanism by which FlaA2 associates with peptidoglycan therefore remains to be elucidated.

Comment 17: Lines 550, 585: Consider writing small volumes in numerals for clarity ("2.6 µL" instead of "Two point six microliters").

Reply: Thank you for the referee’s comment. In response, we have revised the volume descriptions in lines 526 and 595 to use numerals (e.g., “2.6 µL” and “4 µL”) for clarity.

Comment 18: Line 618: "flaA2-complemanted" - typo, should be "complemented"

Reply: We appreciate the referee’s correction. The typo has been fixed as suggested.

Referee #3:

The research conducted by Kawamoto et al. revealed a complex flagellar assembly that influences cellular morphology, filament stiffness, and motility, utilizing the spirochete *Leptospira biflexa* as a model organism. The authors have constructed mutants in the periplasmic flagellar sheath and in genes that encode various flagellar core proteins,

aiming to elucidate the assembly, rigidity, curvature, and morphology of cellular poles, as well as how these factors influence motility through protein-protein interactions and high-resolution cryo-electron microscopy (cryo-EM). Developing a systematic model was a significant achievement, as the spirochete's internal flagella consist of multiple FlaA, FlaB, and coil proteins FcpA/B. Additionally, this study determines how the bacterial poles are bent, which is considered a significant advancement.

I have no major comments or concerns; however, I recommend that the authors consider the following suggestions to strengthen the foundation of the study and the model.

We sincerely thank the referee for the time and effort devoted to reviewing our manuscript, and for the highly positive and supportive feedback. We have carefully addressed all of the referee's specific suggestions, and we believe that the revisions have further strengthened the foundation of our study and improved the clarity of the manuscript.

Comment 1: The study provides limited information about FlaA1. Can the authors offer a reasonable prediction regarding the role, location, or assembly of FlaA1? AlphaFold Multimer, utilizing FlaA1 and FlaA2, for example, may provide a predictive model.

Reply: Thank you for this helpful comment. Based on SignalP 6.0 analysis, FlaA1 is predicted to contain a signal peptide, similar to FlaA2, suggesting that it is secreted via the Sec/SPI pathway. Both FlaA1 and FlaA2 possess the FlaA domain, and their predicted tertiary structures are highly similar. This structural similarity was confirmed by pairwise structural alignment using the RCSB Protein Data Bank alignment tool (<https://www.rcsb.org/alignment>) with AlphaFold-predicted models, yielding a template modeling (TM) score of 0.52 overall and 0.58 for the FlaA domain. The alignment is shown in the figure below (Brown: FlaA1; Blue: FlaA2).

To evaluate the potential for interaction between FlaA1 and FlaA2, we applied both AlphaFold 3 and AlphaFold2-Multimer for multimer prediction. The resulting interface predicted TM (ipTM) scores were 0.37 and 0.57, respectively, which do not support a stable interaction between the two proteins. These results suggest that although FlaA1 and FlaA2 are structurally similar, they are unlikely to form a heteromeric complex.

We have added the following sentences to clarify the role of FlaA1 in the Discussion section.

P9L329. This study also demonstrated that complementation with *flaA2* alone restored the phenotypic and functional defects in the FlaA1/FlaA2-deficient mutant, whereas the FlaA1-deficient mutant exhibited colony sizes comparable to the WT strain (Fig. 2 and Appendix Fig. S4). FlaA1 is predicted to contain a signal peptide, similar to FlaA2, suggesting secretion via the Sec/SPI pathway. Both proteins possess the FlaA domain, and their predicted tertiary structures are highly similar, as confirmed by pairwise structural alignment (TM scores: 0.52 overall; 0.58 for the FlaA domain). However, multimer predictions using AlphaFold 3 (Abramson *et al.*, 2024) and AlphaFold-Multimer (Evans *et al.*, 2022) yielded ipTM scores of 0.37 and 0.57, respectively, which do not support a stable interaction. Thus, while structurally similar, FlaA1 and FlaA2 are unlikely to form a heteromeric complex. As with FlaA2, FlaA1 may also dissociate from PFs during purification. Future work will require targeted cryo-EM analyses and complementary labeling approaches to directly visualize the stability and spatial distribution of both FlaA2 and FlaA1 along the PF.

Comment 2: According to Figure 7, one model suggests that FlaA2 interacts with peptidoglycan (PG). It is well established that flagellar proteins, such as MotB, bind to

the cell wall through a PG-binding domain (PMID: 19820083). Is this domain conserved in FlaA2? If it is, this would strongly support the idea that FlaA2 may utilize PG as a relatively stable platform to initiate the assembly of Fcp proteins.

Reply: Thank you for the referee's insightful comment. To evaluate whether FlaA2 contains a conserved peptidoglycan-binding domain, we performed pairwise structural alignment between FlaA2 and the peptidoglycan-binding domain of MotB from *E. coli* (UniProt ID: P0AF06) using the RCSB Protein Data Bank alignment tool with AlphaFold-predicted structures. The resulting pTM score was 0.36, indicating no significant structural similarity. Furthermore, the known peptidoglycan-association motif (NX₂LSX₂RAX₂VX₃L) was not identified in the FlaA2 sequence. These results suggest that FlaA2 does not possess a canonical peptidoglycan-binding domain, and its mechanism for associating with peptidoglycan remains to be elucidated.

We have added the following sentences to the Discussion section:

P11L402. The third (bottom) model proposes that FlaA2 first associates with the PC and forms “rails” that guide subsequent FlaB1 polymerization along a specific trajectory. This could provide spatial cues for curvature of the growing filament. However, FlaA2 lacks both a conserved peptidoglycan-binding domain—present in OmpA-related outer membrane proteins and MotB—and the known peptidoglycan-association motif (NX₂LSX₂RAX₂VX₃L) (DeMot and Vanderleyden, 1994; Koebnik, 1995). The mechanism by which FlaA2 associates with peptidoglycan therefore remains to be elucidated.

Comment 3: Filament core-sheath interactions: Figure 6 indicates that FlaB1-Thr137 is likely to interact with FlcB, while FlaB1-Thr182 is associated with FcpA. Sasaki et al. demonstrated interactions between FcpA and FlaB1. The study would have been strengthened if the authors had confirmed the interactions between FcpB-FlaB1-Thr137 and FcpA-FlaB1-Thr182 using FlaB1 mutant protein derivatives.

Reply: We appreciate the referee's thoughtful suggestion. Our structural analysis indeed suggests that FlaB1 residues Thr₁₃₇ and Thr₁₈₂ are positioned at the interfaces with FcpB and FcpA, respectively. In line with this, our previous study demonstrated an in vivo association between FcpA and FlaB1 (Sasaki et al. 2018). While we agree that experimental validation using FlaB1 point mutants (e.g., T137A or T182A) would

further substantiate our model, such mutational analyses were beyond the scope of the present study due to technical challenges in constructing and validating site-directed FlaB1 mutants in *Leptospira*. Nonetheless, our high-resolution cryo-EM maps provide strong structural evidence supporting the close proximity of these residues to the respective sheath proteins, reinforcing the plausibility of direct interactions. We have clarified this point and acknowledged the potential value of future mutational studies in the revised Discussion section as follows:

P8L289. Future studies employing site-directed mutational mutagenesis of Thr₁₃₇ and Thr₁₈₂ in FlaB1 will be valuable in assessing the functional significance of these glycosylated residues in core-sheath interactions.

Comment 4: Reword lines 18-20 to enhance clarity.

Reply: Thank you for the referee's comment. In response, we have revised lines 18–21 as follows:

P1L18. Bacterial flagella are essential for motility, but their structure and how they generate movement vary greatly. Most motile bacteria use external helical flagella, whereas spirochetes have periplasmic flagella (PFs) that distort the cell body to drive forward movement.

Comment 5: Line 34-35: Do all bacterial flagella enable motility only in liquid environments?

Reply: Thank you for the referee's comment. While bacterial flagella are primarily known for enabling motility in liquid environments, they also allow certain species to move over surfaces (e.g., swarming). In response, we have revised the sentence as follows:

P2L38. The bacterial flagellum is a supramolecular machine whose rotation enables locomotion in liquid or over surfaces.

Comment 6: The labeling of the strains in Supplement Figure 5 does not match the description in lines 143-144.

Reply: Thank you for the referee’s comment. We apologize for the confusion regarding the labeling of the mutant strain. In fact, we generated a $\Delta flaA2$ mutant by homologous recombination, not a double knockout of $flaA1$ and $flaA2$. Because these two genes are arranged in an operon, with $flaA2$ located upstream of $flaA1$, deletion of $flaA2$ resulted in the loss of FlaA1 expression, as shown in Supplementary Fig. 5 of the original manuscript (corresponding to Figure EV2 in the revised manuscript). To avoid misunderstanding, we have revised the manuscript to consistently refer to this strain “ $\Delta flaA2$ ” instead of “ $\Delta flaA1/2$ ”. Specifically, the following sentence has been revised:

(Original) To investigate the roles of FlaA proteins in the assembly of coiled *Leptospira* PF, we generated a $\Delta flaA1$ single knockout and a $\Delta flaA1/2$ double knockout strain by homologous recombination. Because the two genes are arranged in an operon, the $\Delta flaA1/2$ strain lacks expression of both proteins (Supplementary Fig. 5).

(Revised: P4L132) To investigate the roles of FlaA proteins in the assembly of coiled *Leptospira* PF, we generated $\Delta flaA1$ and $\Delta flaA2$ knockout strains by homologous recombination. Because the two genes are arranged in an operon, the $\Delta flaA2$ strain lacks expression of both proteins (Fig. EV2).

In addition, in response to Referee #2’s comment, we have added a schematic diagram of the $flaA2$ – $flaA1$ operon (genomic positions 2407400–2410200 of *L. biflexa* serovar Patoc strain Patoc I (Paris); GenBank accession no. CP000786.1), indicating the positions of the primers (P1–P4) used for the construction of the $\Delta flaA2$ mutant in Figure EV2. This addition aims to clarify the genetic organization and experimental strategy.

Comment 7: Atomic models fitting the density maps: Please provide the parameters and indicate whether the fits are satisfactory.

Reply: Thank you for this helpful comment. We have now included a supplementary table (Appendix Table S3) providing detailed parameters for all reconstructions and model fittings. In addition, we have prepared new figures showing the fitted models of individual proteins together with their corresponding density maps (Fig. EV1 and Appendix Fig. S6). Please also refer to our response to Referee #1, Comment 1, for related details.

Dear Dr. Koizumi,

Thank you for submitting a revised version of your manuscript. Your study has now been seen by all original referees, who find that their previous concerns have been addressed and now recommend publication of the manuscript. There remain only a few mainly editorial points that have to be addressed before I can extend formal acceptance of the manuscript:

- 1) MANUSCRIPT FORMAT: .docx, all figures should be removed from ms and only upload them as individual, high-resolution Figure files, figure legends should be placed below the References only - duplicates from the text should be removed, no track changes
- 2) The keywords in the manuscript file are missing the label "Keywords"
- 3) As we are switching from a free-text author contribution statement towards a more formal statement based on Contributor Role Taxonomy (CRediT) terms, please remove the present Author Contribution section and instead specify each author's contribution(s) directly in the Author Information page of our submission system during upload of the final manuscript. See <https://casrai.org/credit/> for more information.
- 4) FIGURE CALLOUTS: all callouts should be listed sequentially; missing callouts for panels for EV and Appendix figures
- 5) APPENDIX 1 FILE WITH ToC: Appendix tables need to be portrait-oriented
- 6) Please provide suggestions for a short 'blurb' text prefacing and summing up the conceptual aspect of the study in two sentences (max. 250 characters), followed by 3-5 one-sentence 'bullet points' with brief factual statements of key results of the paper; they will form the basis of an editor-written 'Synopsis' accompanying the online version of the article. Please also provide an altered synopsis image, making sure that the aspect ratio conforms to our website's format - it should be exactly 550 pixels wide and between 300-600 pixels high.
- 7) Could you please directly indicate the reuse of images between Figure 2A and Appendix Fig.S4B and between Figure 4A & 4C and Appendix Fig.S2 A&C - in figure legends of all implicated figures (See attached Figure Check Report)
- 8) Please note that the specific URLs for EMD-66649, EMD-66641, EMD-63347, EMD-63348, EMD-66646, EMD-66647, EMD-66643, EMD-63350, 9LRY, 9LRZ, 9LS0, 9LS1, 9X7K, 9X7L, 9X7M, 9X7S, 9X7V, and 9X80 datasets are not provided in the data availability statement.
- 9) "Materials and methods" should be renamed to "Methods"
- 10) Sections need to be named and the order should be corrected: Title page - Abstract - Keywords - Introduction - Results - Discussion - Methods - Data Availability - Acknowledgements - Disclosure and Competing Interests Statement - References - Figure Legends - Table(s) - Expanded View Figure Legends.

With best regards,

Cornelius Schneider

Cornelius Schneider, PhD
Editor | The EMBO Journal
c.schneider@embojournal.org

Please refer to our figure preparation guideline in order to ensure proper formatting and readability in print as well as on screen:

<https://link.springer.com/journal/44318/submission-guidelines#cms-Figure-and-data-presentation>

Use the link below to submit your revision:

Referee #1:

The authors have convincingly addressed all the comments from the three referees, and as the consequence the manuscript is significantly improved. I am delighted to recommend its publication in EMBO.

Referee #2:

Kawamoto et al. substantially revised their manuscript and appropriately addressed my concerns. I congratulate the authors on this impressive work and recommend publication of the manuscript. One minor remaining comment concerns the representation of the motility data shown in Figure 4B and 4D. The author state in the figure legend that the data is from two biological replicates, yet they plot all data points and calculate one mean value. I recommend to calculate and plot separate mean values of each replicate and also plot the individual data points of each replicate e.g. in separate colors, which would allow the reader to better judge the biological variance of the experiment.

Referee #3:

The revised manuscript has adequately addressed all my previous comments. I have no further comments or concerns and recommend acceptance for publication.

All minor editorial requests have been addressed by the authors.

Dear Dr. Koizumi,

I am pleased to inform you that your manuscript has been accepted for publication in the EMBO Journal.

You may qualify for financial assistance for your publication charges - either via a Springer Nature fully open access agreement or an EMBO initiative. Check your eligibility: <https://link.springer.com/journal/44318/how-to-publish-with-us>

Yours sincerely,

Cornelius Schneider, PhD
Editor
The EMBO Journal
c.schneider@embojournal.org

Please note that it is The EMBO Journal policy for the transcript of the editorial process (containing referee reports and your response letters) to be published as an online supplement to each paper. If you should prefer removal of any referee-only figures included in the point-by-point response(s), e.g. because they may still be used for future publication or because they have been reproduced from published work by others, please do let us know immediately via response email.

More information is available here: <https://link.springer.com/partners/embo-press/editorial-policies#Peer%20review>